# MNO: Multiscale Neural Operator for 3D Computational Fluid Dynamics

## Abstract

Neural operators have emerged as a powerful data-driven paradigm for solving partial differential equations (PDEs), while their accuracy and scalability are still limited, particularly on irregular domains where fluid flows exhibit rich multiscale structures. In this work, we introduce the Multiscale Neural Operator (MNO), a new architecture for computational fluid dynamics (CFD) on 3D unstructured point clouds. MNO explicitly decomposes information across three scales: a global dimension-shrinkage attention module for long-range dependencies, a local graph attention module for neighborhood-level interactions, and a micro point-wise attention module for fine-grained details. This design preserves multiscale inductive biases while remaining computationally efficient. We evaluate MNO on diverse benchmarks, covering steady-state and unsteady flow scenarios with up to 300k points. Across all tasks, MNO consistently outperforms state-of-the-art baselines, reducing prediction errors by 5% to 50%. The results highlight the importance of explicit multiscale design for neural operators and establish MNO as a scalable framework for learning complex fluid dynamics on irregular domains. Code is released at https://anonymous.4open.science/r/MNO-4104.

## 1. Introduction

Neural operators (Lu et al., 2021), as a data-driven approach for solving partial differential equations (PDEs), have attracted increasing attention in accelerating computational fluid dynamics (CFD) (Lin et al., 2009). They provide approximate solutions within seconds (Sun et al., 2024), achieving inference speeds orders of magnitude faster than

traditional numerical methods, e.g., FEM or FVM, enabling real-time computation and design exploration for complex fluid dynamics tasks.

Despite this remarkable efficiency, neural operators still fall short of traditional solvers in accuracy (typically $10^{-3}$ versus $10^{-7}$ relative error). Recent works have sought to close this gap through carefully-designed feature transformations, including spectral mappings FNO (Li et al., 2021), global latent space learning LNO (Wang & Wang, 2024), and Transformer-stacked approach Transolver (Wu et al., 2024), etc. Yet the intrinsic multiscale nature of fluid flow remains (Rahman et al., 2023; Wen et al., 2022) largely underexplored in architectural design, particularly on irregular and unstructured domains. In addition, sole reliance on global modeling often sacrifices local details, while fine-grained attention mechanisms incur prohibitive computational costs, or lead out-of-memory even in a high-end GPU. These challenges highlight the need for architectures that explicitly disentangle and integrate information across multiple spatial scales with a balanced computational budgets.

In this work, we propose a Multiscale Neural Operator (MNO) to tackle typical CFD tasks on irregular domains. The motivation stems from the observation that physical quantities in flow fields exhibit strong multiscale effects: large-scale global trends, localized interactions near object surfaces, and fine-grained pointwise variations. Our goal is to develop a general framework that can faithfully represent objects in 3D flow fields and accurately predict critical physical quantities such as pressure and velocity.

At the core of MNO is a sequence of three-scale blocks, each combining three complementary, parallel modules: (1) a **Global Dimension-Shrinkage Attention** module, which projects $N$ points into a compact set of $M$ modes to capture long-range dependencies; (2) a **Local Graph Attention** module, which adopts a globally shared graph structure together with differential attention to encode $k$-nearest-neighbor interactions, thereby modeling mid-scale neighborhood dynamics; and (3) a **Micro Point-wise Attention** module, which evolves each point's features independently to retain high-frequency variations. The outputs of these modules are fused after each block, enabling MNO to integrate receptive fields across scales and capture a broad spectrum of physical phenomena. Built directly on 3D point

[1]Anonymous Institution, Anonymous City, Anonymous Region, Anonymous Country. Correspondence to: Anonymous Author <anon.email@domain.com>.

Preliminary work. Under review by the International Conference on Machine Learning (ICML). Do not distribute.

clouds, this design avoids mesh constraints and provides a unified framework for extracting global, local, and fine-grained flow representations.

We validate MNO across multiple benchmarks, including wind-tunnel, magnetohydrodynamics, and parachute simulations, covering both steady and unsteady flow regimes with point resolutions ranging from 15K to 300K. Compared to state-of-the-art (SOTA) methods, MNO reduces prediction errors by 5% to 50%, demonstrating consistent improvements in accuracy and robustness on challenging 3D CFD problems. Notably, when modeling complex turbulent spatiotemporal fields, for example Magnetohydrodynamics (MHD), the proposed model leads to accuracy improvements of up to 50%.

In summary, the main contributions are as follows:

- We propose a Multiscale Neural Operator (MNO) that extracts multiscale features directly on the full point cloud graph, rather than through hierarchical resampling. Cross-scale features are tightly coupled within each network block, enabling multiscale interaction at every stage.

- Within each block, MNO introduces three complementary scale-specific operators: a low-rank global operator for long-range dependencies, a differential attention operator for neighborhood-scale interactions, and a point-wise weighted operator for high-frequency local variations.

- MNO is evaluated on five diverse datasets, covering both steady and unsteady CFD and MHD tasks, and show that it consistently outperforms SOTA baselines. In challenging 3D turbulent spatiotemporal modeling scenarios of magnetohydrodynamics, it achieves up to a 50% performance improvement.

## 2. Related Work

Deep learning for PDEs has progressed along two paths: physics-informed networks that enforce PDE constraints during training, and neural operators that learn solution mappings directly from data. We briefly review both directions, emphasizing their use in fluid dynamics and their limitations in multiscale predictions on irregular domains.

### 2.1. Physics-Informed Neural Networks

Physics-Informed Neural Networks (PINNs) (Raissi et al., 2019) embed PDE constraints into the loss function, enabling solution learning without labeled data. Despite inspiring many extensions (Wang et al., 2021; 2022; Karlbauer et al., 2022; Rao et al., 2023), PINNs require task-specific loss design, struggle with unstructured point clouds, and

is hard to scale to high-dimensional or stiff PDEs, limiting their applicability to complex CFD tasks.

### 2.2. Neural Operators

Neural operators learn mappings from initial or boundary conditions, or equation parameters, to PDE solutions in a data-driven manner. Depending on the data representation, existing approaches can be broadly divided into regular-domain and irregular-domain methods.

**Regular Domain Neural Operators** CNO (Raonic et al., 2023) approximates integral operators with convolutional layers, enabling function-to-function mappings on regular grids. FNO (Li et al., 2021) extends this idea by learning PDE operators in Fourier space, efficiently capturing long-range dependencies. AM-FNO (Xiao et al., 2024) further reduces FNO's parameter cost through an amortized kernel that adapts to varying frequency modes. While effective, these models are restricted to structured geometries (e.g., rectangles or cubes) and is hard to transfer to domains with complex or varying shapes.

**Irregular Domain Neural Operators** PointNet (Qi et al., 2017a) and PointNet++ (Qi et al., 2017b) introduce point-based learning with global pooling and hierarchical neighbor search, respectively, though the latter often incurs high cost and may lose fine-scale details. Geo-FNO (Li et al., 2023a) maps irregular meshes into a uniform latent space for FFT-based FNO operations. LNO (Wang & Wang, 2024) encodes point clouds into compact latent tokens and applies Transformer layers for global modeling, while Transolver (Wu et al., 2024) compresses tokens into physical slices for Transformer-based feature extraction. PCNO (Zeng et al., 2025b) combines FNO-style global features with residual and gradient-based local features. HAMLET (Bryutkin et al., 2024) employs a modular input encoder to construct graph transformers for solving PDEs. PhyMPGN (Zeng et al., 2025a) leverages a learnable Laplacian to guide GNNs toward learning within the physically feasible solution space. Despite these advances, most irregular-domain operators emphasize global features, paying limited attention to the coupling between local and global scales.

**Multiscale Neural Operators** U-NO (Rahman et al., 2023) integrates U-Net with neural operators, enabling multiscale PDE mapping. U-FNO (Wen et al., 2022) enhances FNO with local convolutions, while MscaleFNO (You et al., 2024) employs multiple FNO branches to extract features at different scales. These methods improve multiscale representation but remain tied to regular grids, limiting their ability to handle geometric deformations and dynamic flow fields. MGKN (Li et al., 2020) and AMG (Li et al., 2025) achieve multi-scale receptive fields via stage-wise downsampling on irregular geometries. However such downsampling discards important geometric and boundary information. In contrast,

*Figure 1.* (a) The overview of the proposed MNO model with a sequence of three-scale blocks, and (b) A MNO block combining three complementary, parallel modules: (c) a global dimension-shrinkage attention module for long-range dependencies, (d) a local graph attention module for neighborhood-level interactions, and (e) a micro point-wise attention module for fine-grained details.

MNO builds its multi-scale hierarchy on the full flow-field point cloud, with cross-scale features internally coupled at every network stage.

## 3. Method

The proposed Multiscale Neural Operator (MNO) is designed to solve CFD problems directly on unstructured point clouds by integrating global, local, and micro-scale feature learning. The overall architecture follows an Encoder–MNO–Decoder pipeline: the Encoder embeds spatial coordinates and associated attributes of the input points into latent tokens, a sequence of MNO blocks progressively enriches these representations through multiscale attention mechanisms, and the Decoder maps the processed features back to the target physical quantities. This design allows MNO to capture long-range dependencies, neighborhood-level interactions, and fine-grained details simultaneously, providing an efficient and accurate framework for modeling complex fluid dynamics.

In what follows, we first describe the overall model and the input–output format, then introduce the global, local, and micro modules in detail. Finally, we discuss the differences between MNO and existing multiscale approaches for point cloud learning.

### 3.1. Overview of the MNO model

The proposed MNO model, illustrated in Figure 1, is composed of an Encoder, a sequence of MNO blocks, and a Decoder. The input is represented as an array of points, where each point is described by its 3D spatial coordinates and task-specific auxiliary attributes.

The Encoder, implemented as an MLP, embeds these inputs into a latent token space,

$$X = \text{Encoder}(\text{concat}(pos_{in}, feature_{in})), \quad (1)$$

where $pos_{in} \in \mathbb{R}^{N \times 3}$ denotes the 3D coordinates, $feature_{in}$ represents auxiliary features, and $X \in \mathbb{R}^{N \times D}$ are the latent tokens, with $D = 128$ by default. Since the positional information is explicitly included, no additional positional encoding is required.

The latent tokens are then processed by a sequence of MNO blocks, which form the core of the architecture. Each block integrates global, local, and point-wise attention modules to capture multiscale dynamics, progressively enriching the latent representations with hierarchical flow features. Finally, the Decoder, which is the MLP by default, maps the enriched latent features back to the target physical quantities

$$
\begin{aligned}
X_p &= \text{MNO}(X), \\
X_{out} &= \text{Decoder}(X_p),
\end{aligned}
\tag{2}
$$

where $X_p \in \mathbb{R}^{N \times D}$ denotes the processed latent features and $X_{out} \in \mathbb{R}^{N \times O}$ represents the predicted outputs, with $O$ the number of physical variables.

As a concrete example, in the ShapeNet-Car benchmark, after preprocessing (Deng et al., 2024; Wu et al., 2024), the input consists of $N$ points with 3D coordinates $pos_{in}$ and features $feature_{in}$ that include surface normals and signed distance values (Euclidean distance from each air point to the nearest surface point, positive outside the car). This results in an input dimension of $\mathbb{R}^{N \times 7}$. The output $X_{out}$ includes the velocity vector field in the air region and the pressure scalar field on the car surface, with an output dimension of $\mathbb{R}^{N \times 4}$.

### 3.2. Global Dimension-Shrinkage Attention Module

The global module captures long-range dependencies in the 3D discrete domain, enabling the model to extract global patterns such as overall shape and large-scale flow trends.

To encourage this module to capture long-range, low-frequency features, and to address the quadratic computational cost of applying attention on all tokens, we introduce a low rank projection strategy inspired by LNO (Wang & Wang, 2024).

Specifically, the latent features $X \in \mathbb{R}^{N \times D}$ are projected into a compact $M$-dimensional subspace ($M \ll N$) using a learnable projector $P$, then the feature is processed by a multi-head self-attention (MSA) in the reduced space $\mathbb{R}^{M \times D}$, and finally mapped back to the point feature space $\mathbb{R}^{N \times D}$ via the learned inverse projection $Q$. Formally, the global feature $X_{global} \in \mathbb{R}^{N \times D}$ is computed by

$$
\begin{aligned}
W &= \text{MLP}(X), \ W \in \mathbb{R}^{N \times M}, \\
P &= \text{Softmax}_N(W), \ P \in \mathbb{R}^{N \times M}, \\
Q &= \text{Softmax}_M(W), \ Q \in \mathbb{R}^{N \times M}, \\
Z_{lr} &= P^T \cdot X, Z_{lr} \in \mathbb{R}^{M \times D}, \\
Z'_{lr} &= \text{MSA}(Z_{lr}), \ Z'_{lr} \in \mathbb{R}^{M \times D}, \\
X_{global} &= Q \cdot Z_{lr}, \ X_{global} \in \mathbb{R}^{N \times D},
\end{aligned}
\tag{3}
$$

where the matrix $W$ represents the shared weight for projection $P$ and inverse-projection $Q$ matrices with $M = 256$

by default. $\text{Softmax}_N(\cdot)$ and $\text{Softmax}_M(\cdot)$ denote the Softmax function along the $N$ and $M$ dimensions, respectively. $Z_{lr}$ is the low rank space projection of $X$; $Z'_{lr}$ denotes the global feature in the low rank space. The $X_{global}$ is the global feature in 3D space. The $P$ and the $Q$ share a weight matrix, ensuring consistent representation between low rank projection and its inverse while reducing computational cost.

This module has a global receptive field via low rank projections to learn global, slowly-varying features. In the reduced space, attention weights can be computed efficiently at complexity $O(M^2 D)$ instead of $O(N^2 D)$, and the overall cost is dominated by the projection step $O(MND)$. Unlike LNO, which employs a single shared low rank projection throughout the network, MNO introduces an independent low rank projection in each block. This allows multiple low-rank mappings to progressively learn different global feature subspaces layer by layer, enabling different blocks to progressively explore complementary global feature subspaces.

### 3.3. Local Graph Attention Module

The Local Attention module is designed to restrict interactions to geographically nearby points, ensuring that local geometric structures are explicitly preserved. Specifically, a $k$-nearest neighbor (kNN) graph is first constructed using the Euclidean distance of the input 3D coordinates. Each spatial point serves as a graph node, and its $k$ nearest neighbors define the local connectivity.

Inspired by the Point Transformer (Zhao et al., 2021), originally developed for point cloud segmentation, the Local Graph Attention computes neighborhood features for each node by attending only to its $k$ nearest neighbors. The structure of the Local Attention module is illustrated in Figure 1 (d). The local features between the node and its neighboring nodes is computed following.

$$
\begin{aligned}
G_{knn} &= \text{kNN}(pos_{in}), \ G_{knn} \in \mathbb{R}^{N \times k}, \\
Q, K, V &= \text{MLP}(X), \ Q, K, V \in \mathbb{R}^{N \times D}, \\
K_{nbr} &= \text{NNI}(G_{knn}, K), \ K_{nbr} \in \mathbb{R}^{N \times k \times D}, \\
V_{nbr} &= \text{NNI}(G_{knn}, V), \ V_{nbr} \in \mathbb{R}^{N \times k \times D}, \\
P_{nbr} &= \text{NNI}(G_{knn}, pos_{in}), \ P_{nbr} \in \mathbb{R}^{N \times k \times 3}, \\
X_{local} &= \text{DA}(Q_{ctr}, K_{nbr}, V_{nbr}, P_{ctr}, P_{nbr}),
\end{aligned}
\tag{4}
$$

where the function kNN stands for k-nearest neighbor; $pos_{in}$ represents the coordinate position of all nodes; $G_{knn}$ is the shared kNN graph, and NNI denotes the nearest neighbor index based on $G_{knn}$. The $Q_{ctr} = Q$ denotes the features of the center nodes; $K_{nbr}$ and $V_{nbr}$ are the features of neighboring nodes; $P_{ctr} = pos_{in}$ and $P_{nbr}$ represent the original spatial positions of the central nodes and the neighboring nodes, respectively. The function $\text{DA}(\cdot)$ represents the differential attention mechanism.

Unlike prior graph-based approaches (Li et al., 2020; 2025) that reconstruct neighborhood graphs across layers, we share a single kNN graph across all MNO blocks, eliminating repeated neighbor searches and reducing computational overhead. Compared to fixed-radius, kNN enforces a constant neighborhood size, providing predictable memory usage and improved robustness to density variations.

The local neighborhood interactions are captured via differential attention, which is defined as follows.

$$X_{local} = \text{DA}(Q_{ctr}, K_{nbr}, V_{nbr}, P_{ctr}, P_{nbr})$$
$$= \sum_{j=1}^{k} \alpha_j \odot (v_{nbr}^j + \Delta p^j),$$
$$\alpha_j = \frac{exp(\phi_(\Delta x^j + \Delta p^j))}{\sum_{i=1}^{k} exp(\phi_(\Delta x^i + \Delta p^i))}, \quad (5)$$
$$\Delta x^j = q_{ctr} - k_{nbr}^j, \ \Delta x^j \in \mathbb{R}^{N \times D},$$
$$\Delta p^j = \phi_p(p_{ctr} - p_{nbr}^j), \ \Delta p^j \in \mathbb{R}^{N \times D},$$

where the $X_{local} \in \mathbb{R}^{N \times D}$ is the output local features; $j \in [0, k-1]$ denotes the $j$-th neighboring node within the local neighborhood; $\alpha_j \in \mathbb{R}^{N \times D}$ represents the channel-wise attention weight applied to each neighboring node; $\Delta x^j$ denotes the differential feature between the neighboring node and the center node, and $\Delta p^j$ represents the position-enhanced relative displacement between the neighboring node and the center node; $\phi(\cdot)$ denotes an MLP-based similarity relation mapping kernel, while $\phi_p(\cdot)$ represents an explicit spatial displacement mapping implemented by an MLP. The operator $\odot$ denotes element-wise multiplication, and the weighted neighboring features are aggregated by summation along the neighborhood dimension $k$. The overall computational complexity is mainly dominated by the matrix multiplications in the MLPs, resulting in a complexity of $O(NkD^2)$.

For local feature modeling, existing methods primarily rely on discrete convolutions (Raonic et al., 2023) or dot-product self-attention (Li et al., 2025). In contrast, we model local interactions via feature differences between a center node and its neighbors, and parameterize the similarity function using a learnable MLP. Compared to dot-product similarity, feature differences offer greater robustness to spatial shifts and more directly capture local structural variations. The relative positional encodings are further incorporated to explicitly encode geometric relationships within local neighborhoods.

### 3.4. Micro Point-wise Attention Module

The micro-scale features correspond to the intrinsic attributes of individual spatial points. This module implements a point-wise self-attention mechanism, where each token is reweighted solely based on its own feature vector.

As illustrated in Figure 1 (e), token features from the previous block are processed through an MLP followed by a softmax operation to produce point-specific weights, which indicate the relative importance of each token. The scaled features are then combined with the original input via a residual connection

$$X_{micro} = X + Score_p \odot X, \ X \in \mathbb{R}^{N \times D},$$
$$Score_p = \text{Sigmoid}(\text{MLP}(X)), \ Score_p \in \mathbb{R}^{N \times D}, \quad (6)$$

where $X$ denotes the input token features; $Score_p$ represents the point-wise attention weights, and $X_{micro} \in \mathbb{R}^{N \times D}$ is the resulting micro-scale representation. The symbol $\odot$ indicates element-wise multiplication with broadcasting rule across feature dimensions.

The Local Module leverages differential attention to model inter-node relational variations (equation (5)), while the Micro Module restores absolute point-wise states that are annihilated by differential operators. Together, the mid- and high-frequency representations learned by the Local and Micro Modules substantially enhance the Global Module's fine-grained detail perception. The three modules are thus tightly coupled and jointly enable more expressive modeling of physical spatiotemporal flow fields in MNO.

**Remarks on other multiscale models:** Existing multiscale methods for point clouds primarily focus on multi-level sampling operation (Li et al., 2020; 2025; Qi et al., 2017b; Hu et al., 2020). Repeated downsampling and upsampling discard fine-grained geometric details, yielding suboptimal accuracy in flow field prediction. Moreover, neighborhood relationships must be recomputed after each sampling step, incurring substantial computational overhead. The results in Table 6 further confirm this conclusion.

In contrast, our design introduces distinct mechanisms tailored to each scale without resampling. At the global scale, a low rank projection enforces attention to long-range dependencies and low-frequency structures while reducing computational cost. At the local scale, a restricted receptive field ensures that each point interacts only with its nearest neighbors, capturing mid-frequency interactions tied to geometric adjacency. At the micro scale, point-wise modulation refines the representation by recovering high-frequency details. Together, these complementary modules provide a balanced decomposition of global, local, and fine-scale features, enabling accurate and efficient modeling of multiscale dynamics in CFD.

## 4. Experiments

### 4.1. Benchmarks

We evaluate the model performance on five 3D CFD benchmarks, including steady-state flow field benchmarks, Ahmed body (Ahmed et al., 1984; Li et al., 2023b), ShapeNet Car

(Umetani & Bickel, 2018), DrivAerNet++ (Elrefaie et al., 2024), and the unsteady flow field benchmark, Parachute dynamics (Zeng et al., 2025b), Magnetohydrodynamics (MHD) (Ohana et al., 2024).

**Ahmed body** (100k/sample): A vehicle wind tunnel dataset with a bluff-body structure. Inputs consist of the vehicle surface point cloud and auxiliary conditions such as freestream velocity and Reynolds number. The output is the pressure field on the vehicle surface. **Parachute dynamics** (15k/sample): A time-dependent dataset capturing the inflation of parachutes under pressure loads. Inputs include the initial point cloud positions and markers for the umbrella surface and ropes, while outputs are displacement fields at four time steps. **MHD** (260k/sample): A 3D magnetohydrodynamic turbulence dataset. The input comprises observed density, velocity, and magnetic fields from four consecutive time steps, with the goal of predicting all fields at the next time step. **ShapeNet Car** (30k/sample): A car wind tunnel dataset. Inputs include point positions, signed distance values, and surface normals. Outputs are the velocity field in the air region and the pressure field on the car surface. **DrivAerNet++** (300k/sample): A large-scale automotive wind tunnel dataset. Inputs consist of point positions and surface normals, and the output is the pressure field on the car surface. For detailed configurations, please refer to Appendix B.

### 4.2. Comparison on Accuracy

We reproduce several state-of-the-art open-source methods of neural operator for comparative experiments. The training and testing procedures for all baselines are consistent with MNO. The hyperparameters of baselines adhere to their official code repositories or original papers. Detailed settings can be found in Appendix C.

The comparative results are summarized in Table 1 and Table 2. The proposed MNO consistently delivers higher predictive accuracy across all five benchmarks compared to recent baselines. In particular, relative to the current leading methods, Transolver and PCNO, MNO achieves error reductions of 29.51% on the Ahmed Body dataset, 15.82% on Parachute Dynamics, 54.18% on Magnetohydrodynamics, 14.71% on ShapeNetCar, and 4.80% on DrivAerNet++. Notably, on the challenging 3D turbulent spatiotemporal prediction task, MHD, methods based solely on global feature (Geo-FNO, LNO, Transolver) exhibit clear performance degradation, whereas MNO achieves an improvement of about 50%. This can be attributed to the strong nonlinearity and pronounced multi-scale local turbulent structures in MHD, which are difficult to capture with purely global modeling.

Below, we provide an analysis and discussion of some key factors underlying MNO's methodology over state-of-the-

art methods that leading better results. LNO (Wang & Wang, 2024) compresses point clouds into a latent space with limited tokens, where multiple Transformer layers capture global features. This approach resembles our Global Attention module but suffers from noticiable loss of fine-grained details due to heavy compression. Transolver (Wu et al., 2024) employs a global dimension reduction and introduces residual branch between token compression and decompression to reduce information loss. However, it does not explicitly support multiscale feature learning. PCNO (Zeng et al., 2025b) extracts global, gradient, and residual features of the input point cloud. However, its global feature extraction relies on FNO without point cloud compression, limiting scalability for large datasets. Compared to PCNO, MNO provides stronger mid-scale representations through its Local Attention module. AMG (Li et al., 2025) constructs global and local graph structures through multiple downsampling, but this results in significant loss of spatial details, thereby limiting prediction accuracy. Moreover, repeated farthest point sampling (FPS) significantly increases computation time.

### 4.3. Comparison on Model Scales

This experiment aims to demonstrate that, compared with other baselines, the improvements of our MNO model are primarily due to its innovative design of multiscale structure rather than merely an increase in model scale.

To ensure fairness, all models are compared under consistent parameter scales. For our MNO model, the number of blocks is set to 1, 2, 4, and 8 to construct four parameter scales. For Transolver (Wu et al., 2024), the token length is set to 264, and the network depth is adjusted to 1, 2, 4, and 8 to align with the parameter scales. For LNO (Wang & Wang, 2024), the token length is set to 88, 120, 168, and 240 to match the parameter scales. For AMG (Li et al., 2025), the token length is set to 88, 116, 162 and 228 to match the parameter scales.

The experimental results, as shown in Figure 2, indicate that under the same parameter scales, the prediction error of our MNO model is consistently lower than that of the baselines. For example, in Figure 2 (a), our model achieved prediction errors that are 13%-20% lower than the best baseline across the four parameter scales. When the RL2 error is below 0.075, the parameter scale of our model is reduced by 46.61% compared to the best baseline. This indicates that MNO does not need to design redundant parameter spaces for the underlying physical state of the flow field, as adopted by the baselines in Fig. 2.

### 4.4. Attention Modules Ablation Experiments

In each MNO block, the three attention modules are responsible for extracting multiscale features. To better understand

*Table 1.* The comparison results with other methods on Ahmed body and Parachute datasets, in which $RL2_p$ denotes the relative $L_2$ errors (RL2) of the pressure field; $RL2_{x1\sim4}$ represent the RL2 of the displacement field at 4 time steps; $RL2_x$ denotes the total RL2 of 4 time steps; $MAE$ is the mean absolute errors. The subscript "$*$" indicates the result claimed in the original article. The row titled with "Improvement" refers to the degree of advancement compared to the previous best method.

| | Ahmed body | | Parachute | | | | | |
| --- | --- | --- | --- | --- | --- | --- | --- | --- |
| Methods | $RL2_p$ | $MAE_p$ | $RL2_{x1}$ | $RL2_{x2}$ | $RL2_{x3}$ | $RL2_{x4}$ | $RL2_x$ | $MAE_x$ |
| DeepONet (Lu et al., 2021) | 0.3683 | 59.6948 | 1.2620 | 0.7243 | 0.7915 | 0.7667 | 0.7733 | 0.2864 |
| PointNet (Qi et al., 2017a) | 0.1923 | 35.8585 | 0.0955 | 0.0703 | 0.1069 | 0.1427 | 0.1035 | 0.0345 |
| PointNet++ (Qi et al., 2017b) | 0.3366 | 55.5127 | 0.2364 | 0.0923 | 0.1009 | 0.1623 | 0.1165 | 0.0371 |
| Geo-FNO (Li et al., 2023a) | 0.1400 | 26.3723 | 0.0480 | 0.0248 | 0.0353 | 0.0551 | 0.0366 | 0.0114 |
| LNO (Wang & Wang, 2024) | 0.1908 | 30.4570 | 0.0584 | 0.0431 | 0.0484 | 0.0665 | 0.0504 | 0.0147 |
| AMG (Li et al., 2025) | - | - | 0.0432 | 0.0288 | 0.0369 | 0.0539 | 0.0379 | 0.0120 |
| PCNO* (Zeng et al., 2025b) | 0.0682 | - | - | - | - | - | 0.0373 | - |
| PCNO (Zeng et al., 2025b) | 0.0664 | 12.4693 | 0.0238 | 0.0189 | 0.0305 | 0.0515 | 0.0316 | 0.0094 |
| **Ours** | **0.0468** | **7.0465** | **0.0216** | **0.0164** | **0.0259** | **0.0418** | **0.0266** | **0.0081** |
| **Improvement** | **29.51%** | **43.48%** | **9.24%** | **13.23%** | **15.08%** | **18.83%** | **15.82%** | **13.82%** |

*Table 2.* The comparison results with other advanced methods on Magnetohydrodynamics, ShapeNet Car and DrivAerNet++ datasets. $RL2_d$, $RL2_v$, $RL2_m$, and $RL2$ represent the RL2 of density, velocity, magnetic fields, and their average value.

| | Magnetohydrodynamics (MHD) | | | | | ShapeNet Car | | | | DrivAerNet++ | |
| --- | --- | --- | --- | --- | --- | --- | --- | --- | --- | --- | --- |
| Methods | $RL2_d$ | $RL2_v$ | $RL2_m$ | $RL2$ | $MAE$ | $RL2_p$ | $MAE_p$ | $RL2_v$ | $MAE_v$ | $RL2_p$ | $MAE_p$ |
| DeepONet(Lu et al., 2021) | 0.4855 | 1.0004 | 1.0013 | 0.8290 | 0.6193 | 0.4148 | 11.4996 | 0.2075 | 1.2256 | 0.3203 | 27.4931 |
| PointNet(Qi et al., 2017a) | 0.2304 | 0.2771 | 0.3316 | 0.2797 | 0.1855 | 0.0927 | 2.6222 | 0.0314 | 0.1723 | 0.4278 | 42.6893 |
| PointNet++(Qi et al., 2017b) | 0.4343 | 0.6397 | 0.6647 | 0.5795 | 0.4215 | 0.2082 | 5.9648 | 0.0771 | 0.3813 | 0.4617 | 41.8497 |
| Geo-FNO(Li et al., 2023a) | 0.5196 | 1.0043 | 0.9955 | 0.8398 | 0.6255 | 0.1164 | 3.5748 | 0.0737 | 0.4647 | 0.2869 | 26.3732 |
| LNO(Wang & Wang, 2024) | 0.2310 | 0.2675 | 0.3303 | 0.2762 | 0.1826 | 0.0887 | 2.6118 | 0.0267 | 0.1498 | 0.1984 | 18.1088 |
| AMG(Li et al., 2025) | - | - | - | - | - | 0.0770 | 2.0062 | 0.0236 | 0.1203 | - | - |
| Transolver*(Wu et al., 2024) | - | - | - | - | - | 0.0745 | - | 0.0207 | - | - | - |
| Transolver(Wu et al., 2024) | 0.2282 | 0.2655 | 0.3274 | 0.2737 | 0.1804 | 0.0700 | 1.8151 | 0.0230 | 0.1130 | 0.1749 | 15.4372 |
| **Ours** | **0.1031** | **0.1354** | **0.1378** | **0.1254** | **0.08496** | **0.0597** | **1.3796** | **0.0178** | **0.0845** | **0.1665** | **14.6335** |
| **Improve** | **54.82%** | **49.00%** | **57.91%** | **54.18%** | **52.90%** | **14.71%** | **23.99%** | **22.61%** | **25.22%** | **4.80%** | **5.21%** |

their contributions, we conduct ablation studies by selectively enabling different modules.

The results are summarized in Table 3. The ablation visualization of these three attention modules is shown in Appendix A. "Global," "Local," and "Micro" denote using only the corresponding attention module to learn and predict flow fields. "Global+Local" indicates the joint use of both Global and Local Attention modules, "Global+Local+Micro" represents the full MNO block. The "Global+Global+Global" refers to using three identical Global Attention modules to show the improvement stems from the complementary benefits of our multi-scale design, rather than any single type of model. Unless otherwise specified, the number of MNO blocks is fixed at four.

**Performance of independent modules:** Global-only models suffer from severe information loss due to cascaded low-rank projections, resulting in degenerated predictions. By contrast, the Local Attention module achieves the best performance among the three. Local Attention captures

mid-scale features, i.e., mid-frequency information, which is crucial for distinguishing geometric shapes of objects.

**Improvement of global module:** Combining Global and Local Attention substantially improves performance, with relative gains of 63.85%, 80.59%, 26.68%, 10.68%, and 10.73%, across the five datasets compared to using Local Attention alone. This highlights the strong complementarity of global-scale and mid-scale features, showing that their combination captures most of the key physical processes in flow fields.

**Improvement of micro module:** Adding Micro Attention on top of Global+Local yields further improvements of 4.09%, 7.32%, 17.98%, 2.13%, and 2.80%, across the five datasets. The Micro Attention module captures high-frequency variations that serve as fine-scale corrections to mid-frequency features. While its contribution is smaller, it refines predictions and enhances overall accuracy.

**Coupling of multi-scale modules:** The performance of individual modules is limited, while their combination can

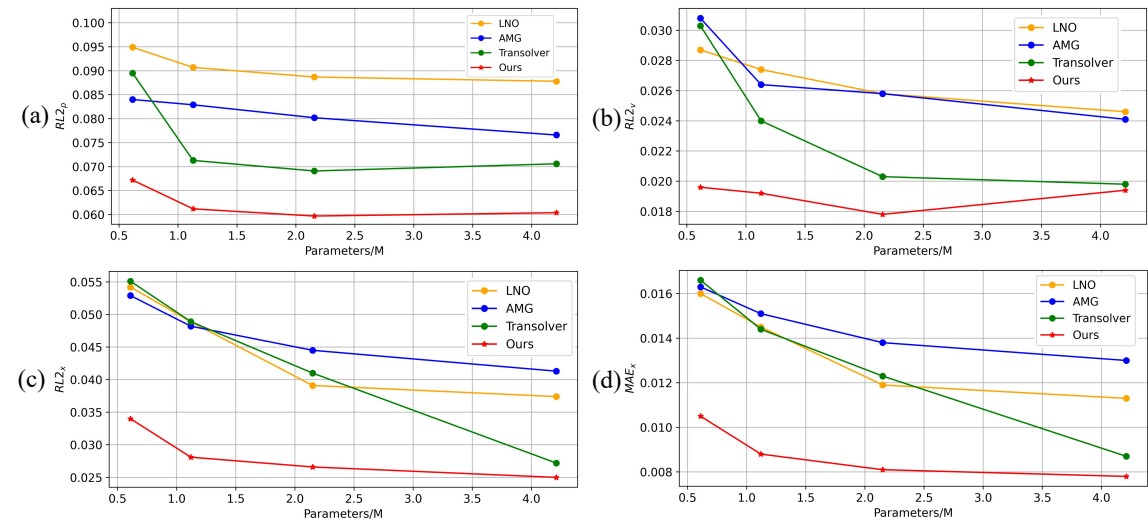

*Figure 2.* Prediction results with identical model parameter counts. (a) the pressure error on ShapeNet Car, (b) the velocity error on ShapeNet Car, (c) the displacement RL2 error on Parachute, and (d) the displacement MAE on Parachute.

*Table 3.* The results of the ablation experiment of Attention Modules. OOM is out of memory.

| Modules | Ahmed body | | Parachute | | MHD | | ShapeNet Car | | | | DrivAerNet++ | |
|---|---|---|---|---|---|---|---|---|---|---|---|---|
| | $RL2_p$ | $MAE_p$ | $RL2_x$ | $MAE_x$ | $RL2$ | $MAE$ | $RL2_p$ | $MAE_{-p}$ | $RL2_v$ | $MAE_v$ | $RL2_p$ | $MAE_p$ |
| Global | 0.8588 | 154.899 | 0.8205 | 0.2629 | 0.7597 | 0.5576 | 0.5117 | 16.9729 | 0.2025 | 1.3195 | 0.7853 | 73.3534 |
| Local | 0.1350 | 24.8293 | 0.1479 | 0.0314 | 0.1712 | 0.1163 | 0.0832 | 2.9235 | 0.0399 | 0.2042 | 0.1919 | 17.8505 |
| Micro | 0.4028 | 64.5137 | 0.2307 | 0.0542 | 0.2728 | 0.1797 | 0.1881 | 6.2470 | 0.0609 | 0.3098 | 0.2396 | 21.2415 |
| Local+Micro | 0.1267 | 24.0813 | 0.1062 | 0.0234 | 0.1344 | 0.0911 | 0.0807 | 2.7964 | 0.0393 | 0.1975 | 0.1908 | 17.6443 |
| Global+Micro | 0.0484 | 7.4700 | 0.0304 | 0.0095 | 0.2722 | 0.1795 | 0.0663 | 1.5408 | 0.0194 | 0.0980 | 0.1728 | 15.3040 |
| Global+Local | 0.0488 | 7.6412 | 0.0287 | 0.0090 | 0.1529 | 0.1041 | 0.0610 | 1.4994 | 0.0201 | 0.0983 | 0.1713 | 14.9961 |
| Global+Global+Global | 0.8591 | 153.907 | 0.8170 | 0.2601 | 0.7603 | 0.5558 | 0.7986 | 23.7095 | 0.3249 | 2.0015 | 0.7853 | 73.2887 |
| Local+Local+Local | OOM | OOM | 0.1484 | 0.0313 | OOM | OOM | 0.0800 | 2.7651 | 0.0385 | 0.1852 | OOM | OOM |
| Micro+Micro+Micro | 0.4328 | 76.6320 | 0.2249 | 0.0527 | 0.2730 | 0.1799 | 0.1893 | 6.1662 | 0.0591 | 0.2930 | 0.2420 | 21.5583 |
| **Global+Local+Micro** | **0.0468** | **7.0465** | **0.0266** | **0.0081** | **0.1254** | **0.08496** | **0.0597** | **1.3796** | **0.0178** | **0.0845** | **0.1665** | **14.6335** |

produce powerful results, which is the design advantage of MNO. Local and Micro modules provide detailed local spatial information to supplement global modules, while global modules provide a broader perspective to enhance Local and Micro units. Therefore, any combination of them can improve performance. These three modules work together to improve the overall performance of MNO, without the need for each module to handle CFD tasks independently. Conversely, if the Global and Micro modules performed well independently, it would indicate architectural redundancy rather than effective specialization.

**Other experiments**: The computing cost is shown in Appendix D. Ablation on the number of MNO blocks, the $M$, and the $k_{nbr}$ are detailed in Appendix E, F and G. The evaluation of the MNO's adaptive resolution capabilities and geometric generalization are detailed in Appendices H and I. For a comparative visualization of the key local flow fields between MNO and the baseline, refer to Appendix K.

## 5. Conclusions

In this work, we introduced the Multiscale Neural Operator (MNO), a new framework for solving CFD problems directly on unstructured point clouds. By explicitly decomposing information into global, local, and micro scales, MNO captures long-range dependencies, neighborhood interactions, and fine-grained details within a unified architecture. Besides performance gains, the ablation and visualization studies confirm the complementary roles of the three attention modules and validate the importance of explicit multiscale design. These results highlight the potential of MNO as a general and efficient framework for learning complex fluid dynamics on irregular domains, paving the way for broader applications of neural operators in large-scale scientific computing.

## Impact Statement

This paper aims to accelerate high fidelity industrial simulations, such as those for automobiles, aircraft, and wind turbines, via multi scale neural operators. Our work may

have various potential societal impacts; however, we believe that none require particular emphasis here.

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

# A. Visualization of Attention Modules

Figure 3 visualizes the prediction errors for different attention configurations. Each row corresponds to one of the five benchmarks, while columns represent the module combinations: the first column shows predictions using only Global Attention; the second column shows Global+Local Attention; the third column repeats the second but with a different color scale for better contrast; and the fourth column shows the full combination of Global, Local, and Micro Attention. The following discussion takes Figure 3 (b) as an illustrative example.

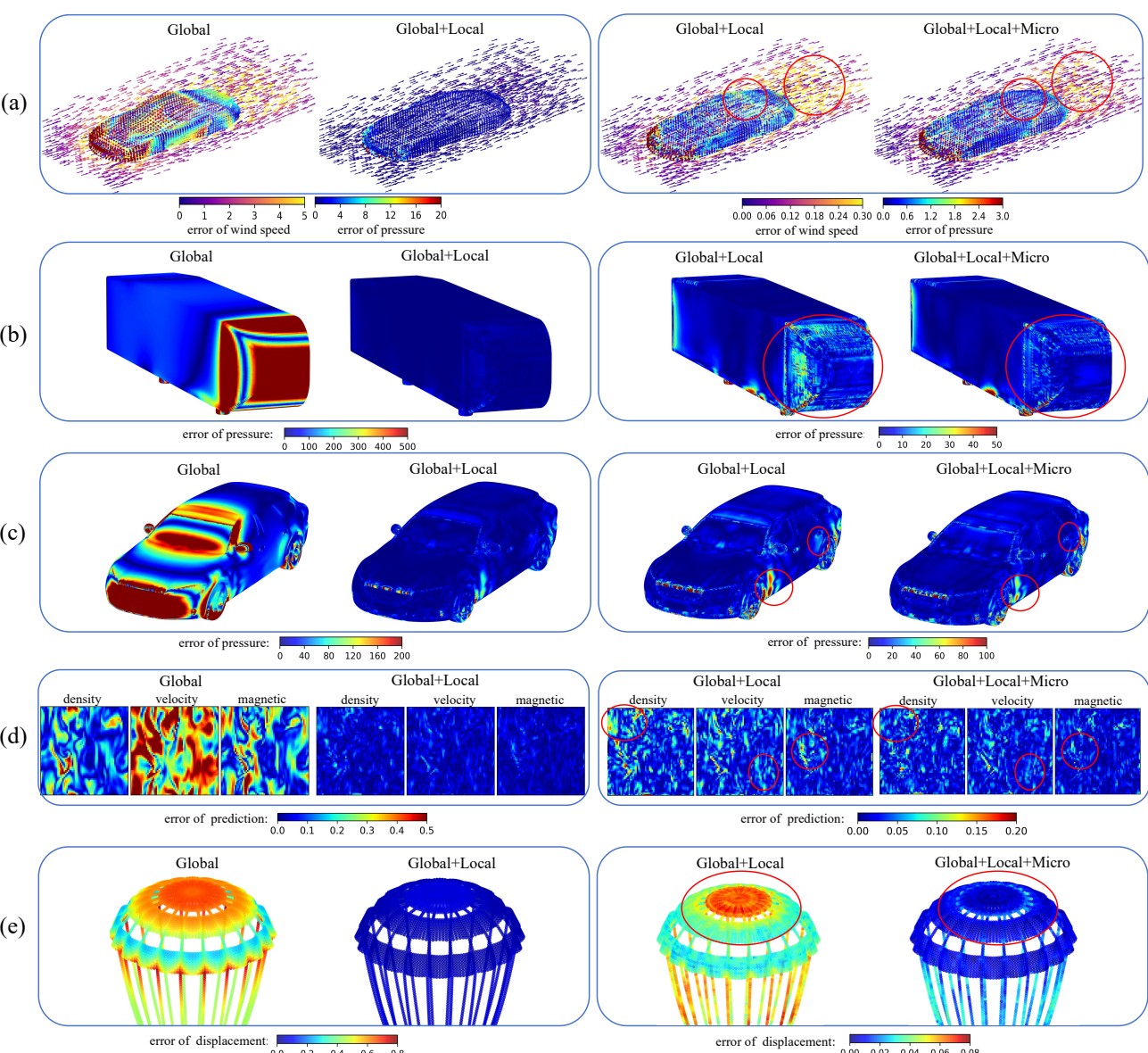

*Figure 3.* The visualization of Global, Local and Micro Attention modules. The red circle serves as a reference for areas with obvious differences. Rows show the error maps for different benchmarks. For each row: (a) ShapeNet Car. The arrow represents the direction of the wind, and the color denotes the prediction error; (b) Ahmed body; (c) DrivAerNet++; (d) Magnetohydrodynamics. All fields are cross sections at the z-axis center; (e) Parachute. For each column: (first column) prediction of only Global Attention module; (second column) prediction of Global and Local Attention modules together; (third column) identical values to the second column but with a different color scale; (fourth column) prediction of the full MNO with Global, Local, and Micro Attention modules together.

From the error map of Global Attention alone, show in the first column of Figure 3 (b), we observe that the module tends to minimize error in the larger side area of the car (the non-windward region). This reflects its capacity to capture low-frequency components: the non-windward region is subject to simpler forces and smaller pressure fluctuations, making

it easier to approximate. In contrast, the windward region experiences stronger forces and larger fluctuations, resulting in higher prediction error.

Comparing Global with Global+Local, it is evident that Local Attention significantly improves performance in the windward region. Local Attention captures mid-frequency information and effectively distinguishes between windward and non-windward regions, complementing the Global Attention module.

Finally, comparing Global+Local with Global+Local+Micro shows that errors in transitional areas between the front and side regions are further reduced when Micro Attention is included. By refining predictions at specific points, Micro Attention supplements fine-grained details and corrects residual errors, demonstrating its role as a complementary high-frequency module.

## B. Details of Benchmarks

This paper conducts a comprehensive evaluation of the model across five benchmarks. The details for each benchmark are provided below.

The Magnetohydrodynamics (MHD) (Ohana et al., 2024) is designed for astrophysical and plasma turbulence research. The ideal MHD equations are solved using a third-order accurate hybrid essentially non-oscillatory (ENO) scheme, with periodic boundary conditions, an isothermal equation of state, and random large-scale solenoidal forcing. Key parameters include the sonic Mach number ($M_S = 0.5, 0.7, 1.5, 2.0, 7.0$) and Alfvén Mach number ($M_A = 0.7, 2.0$). The dataset includes 3D data of density (scalar), velocity (vector), and magnetic field (vector), with 78 training and 10 test samples. Each sample has dimensions (time, x, y, z, fields) = (100, 64, 64, 64, 7), where fields: 1 (density) + 3 (velocity) + 3 (magnetic). The goal is to predict all fields at the next time step using observations from the previous four time steps. Each sample has approximately 260k points in point cloud representation.

The ShapeNet Car (Umetani & Bickel, 2018) focuses on wind tunnel experiments for automobiles, a critical stage in automotive industrial design. This dataset contains 889 samples representing different car shapes, used to simulate driving conditions at a speed of 72 km/h. The car shapes are drawn from the "Car" category of ShapeNet (Chang et al., 2015). The surrounding space is discretized into an unstructured grid with 32,186 points, and both the airflow velocity around the car and the pressure on the car surface are recorded. The number of points on the car surface is 3,682. Following the experimental setup in Transolver (Wu et al., 2024), we use 789 samples for training and the remaining 100 samples for testing. The input point cloud of each sample is preprocessed into a combination of point positions, signed distance functions, and normal vectors. A notable difference is that the original dataset contains 96 fixed noisy points on the car surface. After our preprocessing, the point cloud data consists of 29,498 air points and 3,586 car surface points.

The DrivAerNet++ (Elrefaie et al., 2024) is a large-scale, comprehensive benchmark for automotive aerodynamic design, constructed using high-fidelity CFD simulations. It contains over 8,000 distinct car designs, covering various vehicle types, wheel configurations, and chassis layouts. The inflow air velocity is 108 km/h. We only use a subset of surface pressures for the experiment. To maintain sample diversity while improving research efficiency, we randomly select 200 samples for training and 50 samples for testing. Each point cloud sample consists of approximately 600k points, with each point described by its three-dimensional coordinates (x, y, z) and surface normal vectors (ux, uy, uz). Since the dataset was generated with y-axis symmetry, we only use the points with $y > 0$ (300k) to enhance computational efficiency.

The Ahmed Body (Li et al., 2023b) is a wind tunnel dataset for bluff-body vehicles, used to predict the pressure on the vehicle surface. The vehicle shape is based on the benchmark model designed in (Ahmed et al., 1984). The inflow velocity ranges from 10 m/s to 70 m/s, corresponding to Reynolds numbers from $4.35 \times 10^5$ to $6.82 \times 10^6$. The dataset is generated by systematically varying the vehicle's length, width, height, ground clearance, inclination angle, and rear rounding radius, resulting in a total of 551 samples, each containing approximately 100k surface points. Among these, 500 samples are used for training and 51 samples for testing, consistent with the setup in PCNO (Zeng et al., 2025b). The variations in inflow velocity and Reynolds number are shown in Figure 4.

The Parachute Dynamics (Zeng et al., 2025b) captures the inflation process of different parachutes under specific pressure loads. The pressure load increases linearly from 0 to 1000 Pa over the first 0.1 seconds and then remains constant at 1000 Pa. The learning objective is to map the initial parachute shape to the displacement fields at four specific time points during inflation: $t_1 = 0.04$, $t_2 = 0.08$, $t_3 = 0.12$, and $t_4 = 0.16$. These time points characterize the inflation process, where the parachute first rapidly expands under pressure, then over-expands, and finally rebounds. The experimental setup follows

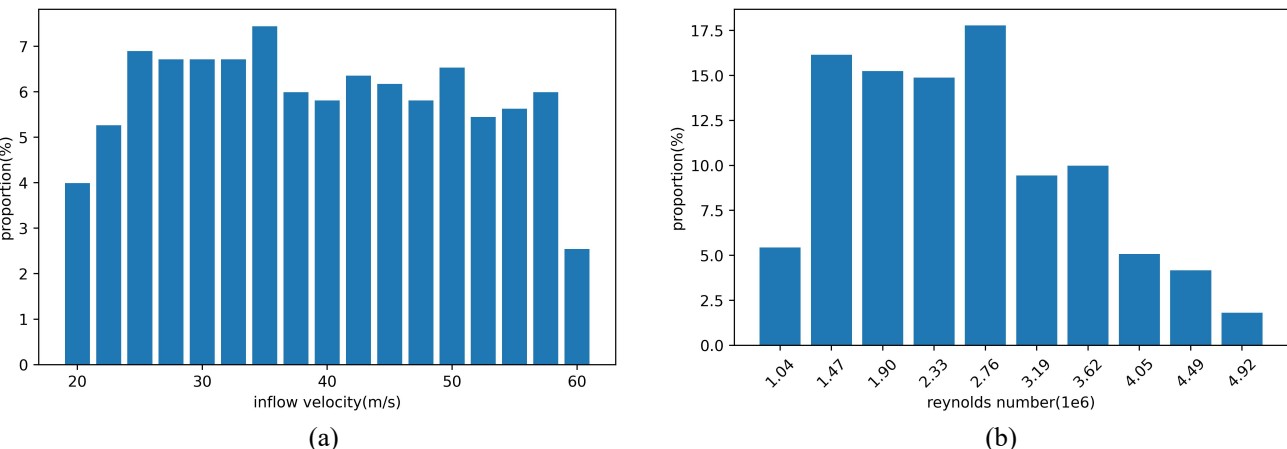

*Figure 4.* Sample distribution of varying inflow velocities and Reynolds numbers for the Ahmed body benchmark case.

that of PCNO (Zeng et al., 2025b), with 1000 samples for training and 200 samples for testing. Each sample contains approximately 15k points in the point cloud.

## C. Full Implementation Details

The implementation software of the model is mainly based on PyTorch 2.4.1, CUDA 12.1, and Python 3.9.0. The computing platform mainly includes Ubuntu 22.04.4 LTS and 4 NVIDIA A800 GPUs.

*Table 4.* The key hyperparameters and training configurations of our MNO and baseline methods.

| Methods | Model key hyperparameters | Training configurations |
|---|---|---|
| DeepONet | branch_dim=128, trunk_dim=128 | |
| PointNet | point numbers=[N,N,N,N,N,N] 
 feature dims= [64,128,1024,512,256,128] | |
| PointNet++ | point numbers = [N,1024, 256, 64, 16, 64, 256, 1024 ,N] 
 feature dims = [32,64, 128, 256, 512, 256, 256, 128, 128] | batch size=4 
 epochs=500 
 Loss=RL2 |
| GeoFNO | modes1=8, modes2=8, modes3=8, width=64, is_mesh=True | Optimizer=AdamW( |
| LNO | n_block=8, n_mode=256, n_dim=128, n_head=8, n_layer=2 | lr=1e-3, weight_decay=5e-5, 
 betas=(0.9, 0.99) ) |
| AMG | feature_width=128, num_layers=3, pos_dim=3, global_ratio=0.25, 
 global_k=4, local_nodes=512, local_ratio=0.25, local_k=6, num_phys=32, 
 num_heads=4 | Scheduler=OneCycleLR( 
 max_lr=1e-3, 
 pct_start=0.2, |
| PCNO | layers=[128, 128, 128, 128, 128], fc_dim=128, 
 Parachute: n_mode=7812, n_measure=2 
 Ahmed body: n_mode=2456, n_measure=1 | div_factor=1e4, 
 final_div_factor=1e4) |
| Transolver | n_hidden=256, n_layers=8, n_head=8, mlp_ratio=2, slice_num=32 | |
| Ours | Block=4, M=256, D=128, Head=8, $k_{nbr}$=16 (8 of MHD and DrivAerNet++) | |

Table 4 details the hyperparameter configurations and training settings of the MNO model and other baseline methods across all benchmarks. Training employed a relative L2 error loss function over 500 epochs, optimized using the AdamW optimizer and the OneCycleLR learning rate scheduler. In the MNO model, $N$ denotes the point number per sample, Block refers to the number of MNO blocks, $M$ indicates the number of tokens in the low rank space of the global attention module, $D$ represents the vector dimension of each token, $Head$ specifies the number of heads in the multi-head self-attention (MSA) mechanism, and $k_{nbr}$ defines the number of neighboring nodes in the local attention module. For MHD and DrivAerNet++, due to GPU memory limitations, $k_{nbr}$ is reduced to 8.

The parameter configurations for all baseline models are derived from the settings provided in the authors' official papers

and code repositories. In Transolver (Wu et al., 2024), n_layers is similar in meaning to Blocks in MNO, and slice_num represents the number of physical slices in the latent space. In LNO (Wang & Wang, 2024), n_block refers to the number of MSA modules in the latent space, and n_mode refers to the number of tokens in latent space. In PointNet++ (Qi et al., 2017b), point numbers refer to the quantity of spatially downsampled or upsampled points at different scale levels, and feature dims refer to the number of channels at each level. This resembles a U-Net-like architecture for point clouds. Other detailed parameter explanations will not be reiterated. Readers can refer to the baselines' original papers and official code repositories for further details.

## D. Computational cost

In this section, we present a detailed analysis of the computational cost of MNO. For a fair comparison, the training and inference times of all models are measured on the same hardware platform (a single A800 GPU with 80 GB memory), with the batch size fixed to 1.

*Table 5.* Statistical results of computational costs for MNO model. $Flops$ and $Params$ represent the model's theoretical computational load and parameter count, respectively. $GPU_t$ and $GPU_i$ denote the training and inference GPU memory per sample, respectively. train time /epoch and inference time /sample stand for the training time per epoch and the inference time per sample, respectively. ">80G" and "OOM" denote out of memory.

| Modules | Flops | Params | $GPU_t$ | $GPU_i$ | train time /epoch | inference time /sample |
|---|---|---|---|---|---|---|
| | | | Ahmed body (100k) | | | |
| Global (no low rank) | 116.6G | 1.16M | >80G | >80G | OOM | OOM |
| Global | 96.5G | 1.49M | 7.1G | 1.0G | 483s | 0.3489s |
| Global+Local | 442.6G | 1.95M | 37.7G | 8.4G | 932s | 1.2054s |
| Global+Local+Micro | 462.5G | 2.15M | 38.6G | 8.4G | 939s | 1.2218s |
| | | | Parachute (15k) | | | |
| Global (no low rank) | 16.759G | 1.16M | >80G | 19.7G | OOM | 0.0946s |
| Global | 13.9G | 1.49M | 1.8G | 0.6G | 120s | 0.0432s |
| Global+Local | 63.7G | 1.95M | 5.9G | 1.67G | 203s | 0.1015s |
| Global+Local+Micro | 66.5G | 2.15M | 6.1G | 1.67G | 205s | 0.0979s |
| | | | Magnetohydrodynamics (260k) | | | |
| Global (no low rank) | OOM | OOM | >80G | >80G | OOM | OOM |
| Global | 252.297G | 1.497M | 17.2G | 2.2G | 185s | 1.0702s |
| Global+Local | 737.091G | 1.963M | 57.3G | 11.8G | 450s | 4.2957s |
| Global+Local+Micro | 788.765G | 2.161M | 59.5G | 11.8G | 453s | 4.3174s |
| | | | ShapeNet Car (30k) | | | |
| Global (no low rank) | 37.2G | 1.16M | >80G | >80G | OOM | OOM |
| Global | 30.9G | 1.49M | 2.7G | 0.7G | 278s | 0.1219s |
| Global+Local | 141.4G | 1.95M | 12.4G | 3.1G | 402s | 0.2917s |
| Global+Local+Micro | 147.8G | 2.15M | 12.7G | 3.1G | 408s | 0.2971s |
| | | | DrivAerNet++ (300k) | | | |
| Global (no low rank) | OOM | OOM | >80G | >80G | OOM | OOM |
| Global | 286.8G | 1.49M | 19.7G | 2.4G | 533s | 1.2204s |
| Global+Local | 841.6G | 1.95M | 65.4G | 13.4G | 1404s | 5.3302s |
| Global+Local+Micro | 900.8G | 2.15M | 67.9G | 13.4G | 1413s | 5.3542s |

In Table 5, "Global (no low rank)" refers to the standard self-attention method, which computes global attention across all spatial points directly without using low rank projection for compression. Compared to this standard self-attention method, the low rank projection reduces the inference time by 54.3%. Furthermore, due to training GPU memory requirements exceeding 80GB, the standard self-attention method could not complete the training task on any benchmark used in this paper.

"Global+Local+Micro" denotes the full MNO model. For physical fields with 15k, 30k, 100k, 260k, and 300k points, MNO requires only 0.09s, 0.3s, 1.2s, 4.3s, and 5.3s for inference, respectively. This demonstrates second-level inference latency, meeting the requirements of large-scale CFD prediction. Moreover, MNO supports multi-sample parallelism, with further efficiency gains achieved by increasing the batch size or the number of GPUs.

*Table 6.* Comparison of computational costs with the latest baseline. "Params" represent the model's parameter count, respectively. "train/epoch" and "inference/sample" are the training time per epoch and the inference time per sample, respectively. $n_{train}$ denotes the training samples of the benchmark. $N$ is the number of points per sample.

| Benchmark | $N$ | $n_{train}$ | Method | Params | train/epoch | inference/sample |
|---|---|---|---|---|---|---|
| Parachute | 15k | 1000 | AMG(Li et al., 2025) | 2.15M | 0.29h | 1.0335s |
| | | | **Ours** | 2.15M | **0.06h** | **0.0979s** |
| ShapeNet Car | 30k | 789 | AMG(Li et al., 2025) | 2.15M | 0.88h | 3.9245s |
| | | | **Ours** | 2.15M | **0.11h** | **0.2971s** |
| Ahmed body | 100k | 500 | AMG(Li et al., 2025) | 2.15M | 4.58h | 32.828s |
| | | | **Ours** | 2.15M | **0.26h** | **1.2218s** |
| Magnetohydrodynamics | 260k | 78 | AMG(Li et al., 2025) | 2.15M | 4.81h | 221.54s |
| | | | **Ours** | 2.15M | **0.13h** | **4.3174s** |
| DrivAerNet++ | 300k | 200 | AMG(Li et al., 2025) | 2.15M | 16.22h | 291.15s |
| | | | **Ours** | 2.15M | **0.39h** | **5.3542s** |

Table 6 compares the computational speed of MNO with the recent SOTA method AMG under comparable parameter budgets. Although both methods adopt graph structures and multi-scale strategies, MNO reduces actual inference time by about 90%. This is because AMG follows a conventional multi-level point-cloud reconstruction strategy, where the computational graph is rebuilt at each stage based on the current features, and adjacency relationships are computed using FPS. In contrast, all MNO blocks share a single graph, so adjacency is computed only once at the frontend, substantially reducing computation. Meanwhile, the shared graph built on original spatial coordinates explicitly enhances geometric representation in the latent space, leading to a 21.1% accuracy improvement compared to AMG (Figure 2(a)).

## E. The Ablation of the Depth of MNO

This experiment aims to explore performance changes in Global Attention, Local Attention, and Micro Attention modules with varying depths of the MNO model.

Due to the large number of models requiring training in this experiment, to enhance experimental efficiency, ablation studies are performed exclusively on the smaller-scale point cloud datasets: ShapeNet Car and Parachute. ShapeNet Car necessitates simultaneous prediction of velocity and pressure fields, while Parachute incorporates temporal information, making both highly representative benchmarks.

Table 7 presents the experimental results. It is evident that the MNO model incorporating all three attention modules achieves the highest prediction accuracy in most cases. A significant improvement in MNO's predictive performance is observed as the number of blocks increases from 1 to 4. However, performance gains become marginal when the block count exceeds 4, suggesting that the model likely enters a saturated state at this stage.

MNO achieves satisfactory performance with only four serially connected blocks. Because that the kNN graph–based Local Module provides strong inductive bias and tends to saturate with relatively shallow depth, unlike deep Transformers that rely heavily on stacking for expressivity. Figure 2 also confirms this: MNO can achieve better performance than Transolver with fewer block numbers.

*Table 7.* The ablation experimental results of depth of MNO model. Blocks refer to the number of cascaded MNO blocks in the model. $RL2_{x1\sim4}$ represent the RL2 of the displacement field at 4 time steps. $RL2_x$ denotes the total RL2 of 4 time steps.

| Blocks | Modules | ShapeNet Car | | | | Parachute | | | | | |
|---|---|---|---|---|---|---|---|---|---|---|---|
| | | $RL2_v$ | $MAE_v$ | $RL2_p$ | $MAE_p$ | $RL2_{x1}$ | $RL2_{x2}$ | $RL2_{x3}$ | $RL2_{x4}$ | $RL2_x$ | $MAE_x$ |
| 1 | Global | 0.0252 | 0.1315 | 0.0813 | 2.1268 | 0.0535 | 0.0380 | 0.0439 | 0.0615 | 0.0455 | 0.0135 |
| | Local | 0.0526 | 0.2637 | 0.1191 | 4.1156 | 0.0777 | 0.0872 | 0.1263 | 0.2221 | 0.1353 | 0.0315 |
| | Micro | 0.0594 | 0.2925 | 0.1924 | 6.3051 | 0.1942 | 0.1967 | 0.2504 | 0.3138 | 0.2408 | 0.0581 |
| | Global+Local | 0.0220 | 0.1130 | 0.0686 | 1.7701 | 0.0319 | 0.0250 | 0.0358 | 0.0531 | 0.0361 | 0.0110 |
| | Global+Local+Micro | 0.0196 | 0.0999 | 0.0672 | 1.6221 | 0.0310 | 0.0239 | 0.0336 | 0.0500 | 0.0340 | 0.0105 |
| 2 | Global | 0.0266 | 0.1481 | 0.0852 | 2.1712 | 0.0342 | 0.0236 | 0.0323 | 0.0490 | 0.0330 | 0.0102 |
| | Local | 0.0465 | 0.2328 | 0.0959 | 3.4201 | 0.0569 | 0.0641 | 0.1022 | 0.1671 | 0.1045 | 0.0251 |
| | Micro | 0.0586 | 0.0586 | 0.1906 | 0.2893 | 0.1813 | 0.1894 | 0.2429 | 0.3042 | 0.2335 | 0.0553 |
| | Global+Local | 0.0197 | 0.0972 | 0.0632 | 1.5172 | 0.0245 | 0.0185 | 0.0291 | 0.0453 | 0.0294 | 0.0093 |
| | Global+Local+Micro | 0.0192 | 0.0946 | 0.0612 | 1.4408 | 0.0231 | 0.0172 | 0.0276 | 0.0441 | 0.0281 | 0.0088 |
| 4 | Global | 0.2025 | 1.3195 | 0.5117 | 16.9729 | 0.7308 | 0.7884 | 0.8502 | 0.8143 | 0.8205 | 0.2629 |
| | Local | 0.0399 | 0.2042 | 0.0832 | 2.9235 | 0.0455 | 0.0911 | 0.1545 | 0.2099 | 0.1479 | 0.0314 |
| | Micro | 0.0609 | 0.3098 | 0.1881 | 6.2470 | 0.1812 | 0.1869 | 0.2402 | 0.3010 | 0.2307 | 0.0542 |
| | Global+Local | 0.0201 | 0.0983 | 0.0610 | 1.4994 | 0.0250 | 0.0183 | 0.0289 | 0.0438 | 0.0287 | 0.0090 |
| | Global+Local+Micro | 0.0178 | 0.0845 | 0.0597 | 1.3796 | 0.0216 | 0.0164 | 0.0259 | 0.0418 | 0.0266 | 0.0081 |
| 8 | Global | 0.3252 | 1.9870 | 0.7991 | 23.7101 | 0.7659 | 0.8413 | 0.8981 | 0.8552 | 0.8661 | 0.2713 |
| | Local | 0.0319 | 0.1644 | 0.0728 | 2.3717 | 0.1106 | 0.1242 | 0.1916 | 0.2549 | 0.1790 | 0.0472 |
| | Micro | 0.0584 | 0.2916 | 0.1880 | 6.1920 | 0.1727 | 0.1804 | 0.2361 | 0.2949 | 0.2254 | 0.0525 |
| | Global+Local | 0.0201 | 0.1006 | 0.0614 | 1.4902 | 0.0392 | 0.0313 | 0.0401 | 0.0558 | 0.0402 | 0.0123 |
| | Global+Local+Micro | 0.0194 | 0.0857 | 0.0604 | 1.3986 | 0.0239 | 0.0163 | 0.0235 | 0.0399 | 0.0250 | 0.0078 |

Each module operates at distinct receptive field scales: the global module captures domain-level dependencies, while the local and micro modules provide fine-grained neighborhood and point-level information. Conversely, if the global and micro modules function effectively in isolation, it indicates architectural redundancy rather than an effective and specialized model.

## F. The Ablation of $M$ in low rank space

To investigate the size of the model's demand for low rank space representation capacity, we conduct ablation experiments with parameter $M$. Table 8 shows ablation results for $M$. When $M >= 256$, model performance saturates, indicating limited capacity requirements in the low rank space. For CFD tasks with limited computational resources, we recommend reducing $M$, as performance does not drop significantly.

*Table 8.* The low rank space $M$ ablation results on ShapeNet Car.

| Methods | $M$ | $RL2_p$ | $MAE_p$ | $RL2_v$ | $MAE_v$ | $Flops$ | $Params$ | $GPU_t$ | $GPU_i$ |
|---|---|---|---|---|---|---|---|---|---|
| MNO | 8 | 0.0607 | 1.4416 | 0.0190 | 0.0911 | 143.584G | 2.02M | 12.4G | 3.05G |
| MNO | 16 | 0.0606 | 1.4320 | 0.0197 | 0.0917 | 143.720G | 2.03M | 12.4G | 3.05G |
| MNO | 32 | 0.0598 | 1.3803 | 0.0180 | 0.0857 | 143.992G | 2.04M | 12.4G | 3.05G |
| MNO | 64 | 0.0599 | 1.4123 | 0.0187 | 0.0910 | 144.536G | 2.05M | 12.4G | 3.05G |
| MNO | 128 | 0.0607 | 1.4201 | 0.0188 | 0.0888 | 145.625G | 2.08M | 12.6G | 3.05G |
| MNO | 256 | **0.0597** | **1.3796** | 0.0178 | 0.0845 | 147.802G | 2.15M | 12.7G | 3.05G |
| MNO | 512 | 0.0601 | 1.3763 | **0.0177** | **0.0822** | 152.155G | 2.28M | 13.0G | 3.12G |

Increasing the value of $M$ does not lead to significant improvements. This indicates that the low rank space does not rely on a large dimensionality setting. We speculate that among objects of the same category (such as different vehicles in ShapeNet Car), the overall contours exhibit high structural similarity, primarily composed of low-frequency information, while the key factors affecting prediction accuracy depend more on mid- to high-frequency details. Therefore, a smaller low rank space is sufficient to capture the low-frequency features of the overall contours, whereas finer geometric structures are dominated by mid- to high-frequency components. This result suggests that in flow field prediction tasks, low-frequency information

primarily serves an auxiliary global constraint role, while mid- to high-frequency information is more critical for recovering local details.

## G. The Ablation of $k_{nbr}$ of Local Graph attention

To investigate the model's sensitivity to the number of neighbors, an ablation study on the $k_{nbr}$ is conducted. The experimental results are shown in Table 9. When $k >= 16$, the performance of the MNO reaches saturation. The GPU memory is linearly related to the value of k.

*Table 9.* The ablation results of $k_{nbr}$ of Local graph attention on ShapeNet Car. $GPU_t$ and $GPU_i$ represent the training and inference GPU memory per sample, respectively.

| Methods | $k_{nbr}$ | $RL2_p$ | $MAE_p$ | $RL2_v$ | $MAE_v$ | $Flops$ | $Parameters$ | $GPU_t$ | $GPU_i$ |
|---|---|---|---|---|---|---|---|---|---|
| Local | 2 | 0.1902 | 5.8130 | 0.0627 | 0.3134 | 32.987G | 0.83M | 2.9G | 0.97G |
| Local | 4 | 0.1154 | 3.8378 | 0.0488 | 0.2407 | 45.742G | 0.83M | 4.1G | 1.35G |
| Local | 8 | 0.0943 | 3.2675 | 0.0428 | 0.2230 | 71.252G | 0.83M | 6.4G | 1.91G |
| Local | 16 | 0.0832 | 2.9235 | 0.0399 | 0.2042 | 122.272G | 0.83M | 11.2G | 3.05G |
| Local | 32 | **0.0746** | **2.4595** | **0.0350** | **0.1780** | 224.311G | 0.83M | 21.1G | 5.32G |
| MNO | 2 | 0.0648 | 1.4644 | 0.0184 | 0.0860 | 58.517G | 2.15M | 4.7G | 1.02G |
| MNO | 4 | 0.0645 | 1.4378 | 0.0186 | 0.0861 | 71.272G | 2.15M | 5.6G | 1.44G |
| MNO | 8 | 0.0635 | 1.4210 | 0.0183 | 0.0855 | 96.782G | 2.15M | 7.8G | 1.91G |
| MNO | 16 | **0.0597** | **1.3796** | **0.0178** | **0.0845** | 147.802G | 2.15M | 12.7G | 3.05G |
| MNO | 32 | 0.0599 | 1.4160 | 0.0191 | 0.0935 | 249.841G | 2.15M | 22.3G | 5.32G |

When the Local Module independently handles the CFD task, the prediction error rapidly decreases as $k_{nbr}$ increases. However, when the MNO model performs the CFD prediction task, the reduction in prediction error with increasing $k_{nbr}$ is more gradual. The Local Module lacks a global perspective of the point cloud. Increasing $k_{nbr}$ effectively expands its receptive field, making the Local Module highly sensitive to changes in $k_{nbr}$. Nevertheless, a larger $k_{nbr}$ significantly increases GPU memory usage and computational complexity, so its value cannot be set excessively high. The MNO model possesses receptive fields at multiple scales. The global perspective provided by the Global Module alleviates the Local Module's strong dependence on a large receptive field, allowing the Local Module to focus more on analyzing local features. Consequently, the prediction performance of MNO is less sensitive to variations in $k_{nbr}$.

Since GPU memory is exceptionally sensitive to $k_{nbr}$, we must minimize the number of neighbor nodes $k_{nbr}$ to avoid memory overflow when the dataset contains a large number of sampled points. Fortunately, the unique multi-scale structure design of MNO mitigates the heavy reliance of graph structures on computational resources. As shown in Table 9, when the RL2 error is also less than 0.08, compared to the Local Graph Attention method, MNO reduces computational Flops by 73.9% and GPU memory usage by 83.8%. Therefore, even when $k_{nbr}$ is reduced to prevent memory overflow, MNO can still achieve excellent prediction performance. For instance, in the DrivAerNet++ dataset used in this study, the point cloud scale of 300k forces us to reduce $k_{nbr}$ from 16 to 8, yet the prediction error remains lower than that of the best baseline.

## H. Zero-Shot Resolution Adaptation Study

In this experiment, we designed an adaptive resolution training strategy that enables the MNO model to support point cloud inputs with arbitrarily varying resolutions within a certain range in a zero-shot manner. The training method is illustrated in the following equation:

$$
\begin{aligned}
mask &= \text{randmask}(sample\ rate), \\
X_{in}^{sample} &= X_{in} \odot mask, \\
X_{out}^{sample} &= \text{model}(X_{in}^{sample}), \\
Y^{sample} &= Y \odot mask, \\
L &= \text{Loss}(X_{out}^{sample}, Y^{sample})
\end{aligned}
\tag{7}
$$

where $sample\ rate$ represents the proportion of sampling points, and $sample\ rate \in (10\%, 100\%)$, $mask \in \mathbb{R}^{N_{\max} \times 1}$, and $N_{\max}$ represents the maximum limit of input points. The function randmask($\cdot$) randomly sets $n$ positions in the mask to 1, where $n = \text{round}(N \times sample\ rate)$, and sets the remaining positions to 0. $X_{in} \in \mathbb{R}^{N_{\max} \times f}$ represents the original point cloud input data, where $f$ is the number of input features. $X_{out}^{sample}$ corresponds to the predicted physical field data of the sampled points, $Y$ is the ground truth, and Loss is the loss function.

*Table 10.* Zero-Shot results across different resolutions on DrivAerNet++ benchmark. Sample rate refers to the proportion of spatial sampling points. $RL2_p$, $MAE_p$, $RMSE_p$ and $MSE_p$ denote the Relative L2 error, Mean Absolute Error, Root Mean Square Error, and Mean Square Error of the pressure field, respectively.

| sample rate | $n$ | $RL2_p$ | $MAE_p$ | $RMSE_p$ | $MSE_p$ |
|---|---|---|---|---|---|
| 10% | 30k | 0.1793 | 15.7110 | 27.9917 | 816.6555 |
| 20% | 60k | 0.1727 | 15.1267 | 26.9546 | 743.8958 |
| 30% | 90k | 0.1708 | 15.0152 | 26.6468 | 725.7269 |
| 40% | 120k | 0.1728 | 14.9423 | 27.0181 | 753.2448 |
| 50% | 150k | 0.1729 | 14.9048 | 26.9991 | 759.3806 |
| 60% | 180k | 0.1712 | 14.9021 | 26.7007 | 735.7252 |
| 70% | 210k | 0.1715 | 14.9023 | 26.7680 | 739.0064 |
| 80% | 240k | 0.1725 | 14.9164 | 26.9393 | 754.6640 |
| 90% | 270k | 0.1719 | 14.9203 | 26.8491 | 747.4118 |
| 100% | 300k | 0.1717 | 14.9379 | 26.8093 | 743.3867 |

During model training, the hyperparameter settings are consistent with Table 4. The results are shown in Table 10. Across 10 input resolutions (100% → 10%), MNO maintains stable performance with RL2 = 0.1755 ± 0.0038. This proves that even in severely sparse situations, it exhibits strong robustness and consistent physical state prediction for subsampling and density variations.

# I. The Geometric Generalization Study

In this section, we investigate the geometric generalization capability of the MNO. The model is trained exclusively on one category of geometric shapes and tested on another category to evaluate its performance.

The car shapes in the DrivAerNet++ benchmark are categorized into Sedan and Sport Utility Vehicle (SUV), as illustrated in Figure 5. Sedans typically feature a low-center-of-gravity, streamlined body design that emphasizes road-handling stability and high-speed cruising capability. In contrast, SUVs are characterized by higher ground clearance and a more boxy, taller body, providing a more spacious cargo area and enhanced off-road performance. Accordingly, we conduct ablative experiments to evaluate geometric generalization performance for the two vehicle types.

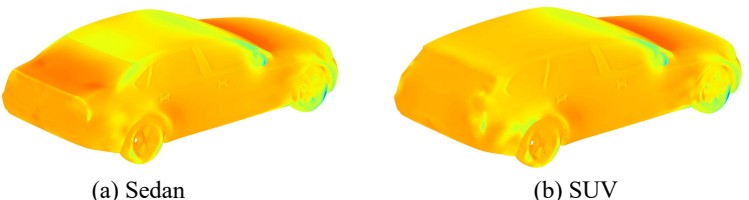

(a) Sedan                    (b) SUV

*Figure 5.* Display of different geometric shapes of Sedan and SUV.

The experimental results are summarized in Table 11. The predicted RL2 errors for all four cases are below 0.22, indicating that the MNO possesses a certain degree of geometric generalization ability. It demonstrates that the model can learn some universal physical laws from training cases of a single geometric type. When the model is trained solely on Sedan-type shapes, the predicted RL2 error is approximately 0.14 for Sedan cases, while the error increases to around 0.20 for SUV cases. The converse also holds true. This suggests that the geometric generalization capability of the MNO is limited. The geometric homogeneity and limited quantity of the training samples cause the model to overfit to some extent.

Increasing the diversity of training samples can effectively alleviate this issue. For example, if the training set of the MNO

*Table 11.* Research results on zero-shot transfer of different geometric shapes.

| Train | Test | $RL2_p$ | $MAE_p$ | $RMSE_p$ | $MSE_p$ | Train samples | Test samples |
|-------|------|---------|---------|----------|---------|---------------|--------------|
| Sedan | Sedan | 0.1468 | 14.0959 | 23.3102 | 173.2603 | 200 | 50 |
| Sedan | SUV | 0.2020 | 19.6659 | 31.2958 | 290.4522 | 200 | 50 |
| SUV | SUV | 0.1419 | 12.4015 | 22.1808 | 162.3201 | 200 | 50 |
| SUV | Sedan | 0.2189 | 20.1559 | 34.6224 | 368.7184 | 200 | 50 |

includes both vehicle types, the RL2 error will decrease from 0.2 to 0.16. In the future, we will collect and construct a large-scale 3D CFD benchmark encompassing a wider variety of geometric types and a larger sample size to enhance the generalization capability of the MNO.

## J. The Display and Discussion of Prediction Results of MNO Model

In this section, we present prediction results obtained by the proposed MNO model, as illustrated in Figures 6, Figure 7, Figure 8, and Figure 10. It is evident that across all datasets, the model's predictions exhibit strong consistency with the ground truth, with prediction errors approaching zero in most regions of the point cloud. These results confirm that the MNO model is capable of capturing the majority of physical behaviors in fluid flows, making it highly suitable for CFD tasks.

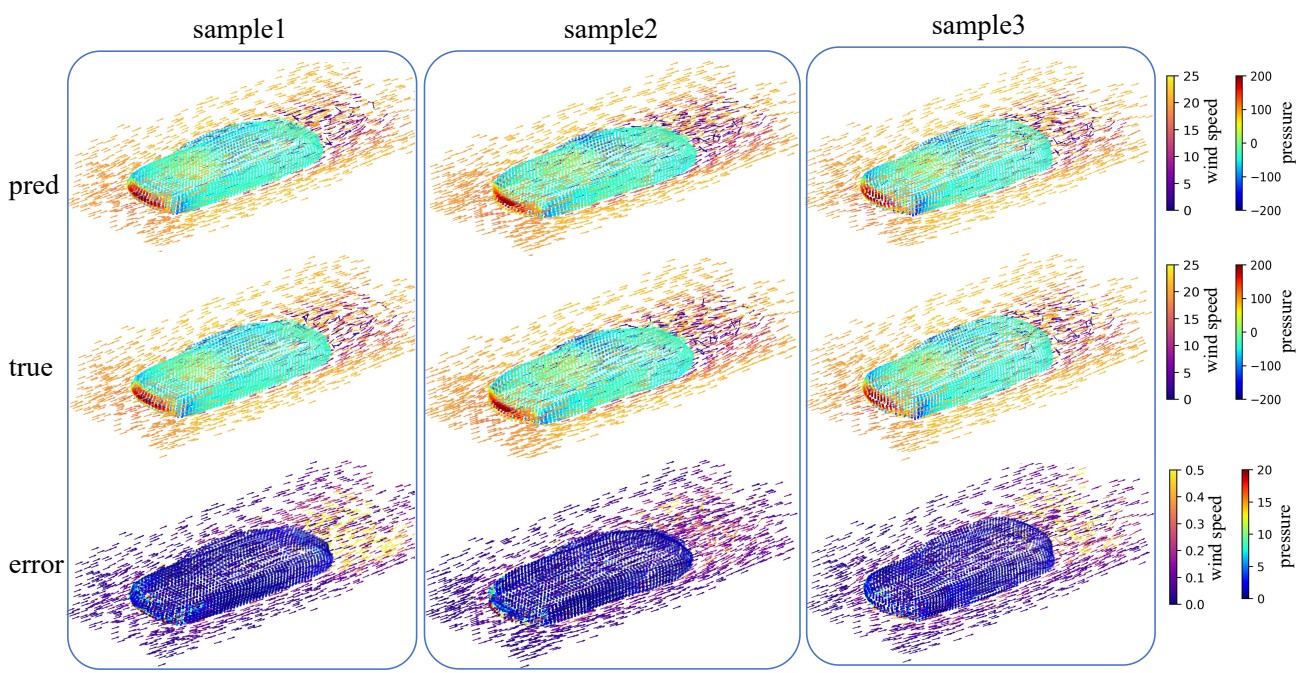

*Figure 6.* The display of prediction results on ShapeNet Car dataset. The pred represents the predicted velocity and pressure fields, the true denotes the ground truth, and the error stands for the absolute error of the prediction fluid fields. The arrows represent the wind direction, and the color of arrows denotes the magnitude of wind speed.

Among the benchmarks, the ShapeNet Car dataset merits particular attention, as the model is required to simultaneously predict both the velocity field of the airflow around the car and the pressure field normal to the car surface. Figure 6 presents the experimental results on this dataset. From the "true" visualization, one can observe that the windward regions of the car surface exhibit higher pressure, while the leeward and side regions experience lower pressure. As the airflow passes the vehicle body, its velocity decreases and complex wake turbulence forms downstream of the car. In the "pred" visualization, the model successfully reproduces the contrast between windward and leeward surfaces, as well as the turbulent structures in the wake, indicating that MNO has effectively learned the underlying PDEs governing wind tunnel phenomena from point cloud data. In the "error" visualization, relatively large prediction errors are observed at the front surface of the car, where the windward face encounters high-speed inflow and rapid flow variations, making the prediction more challenging.

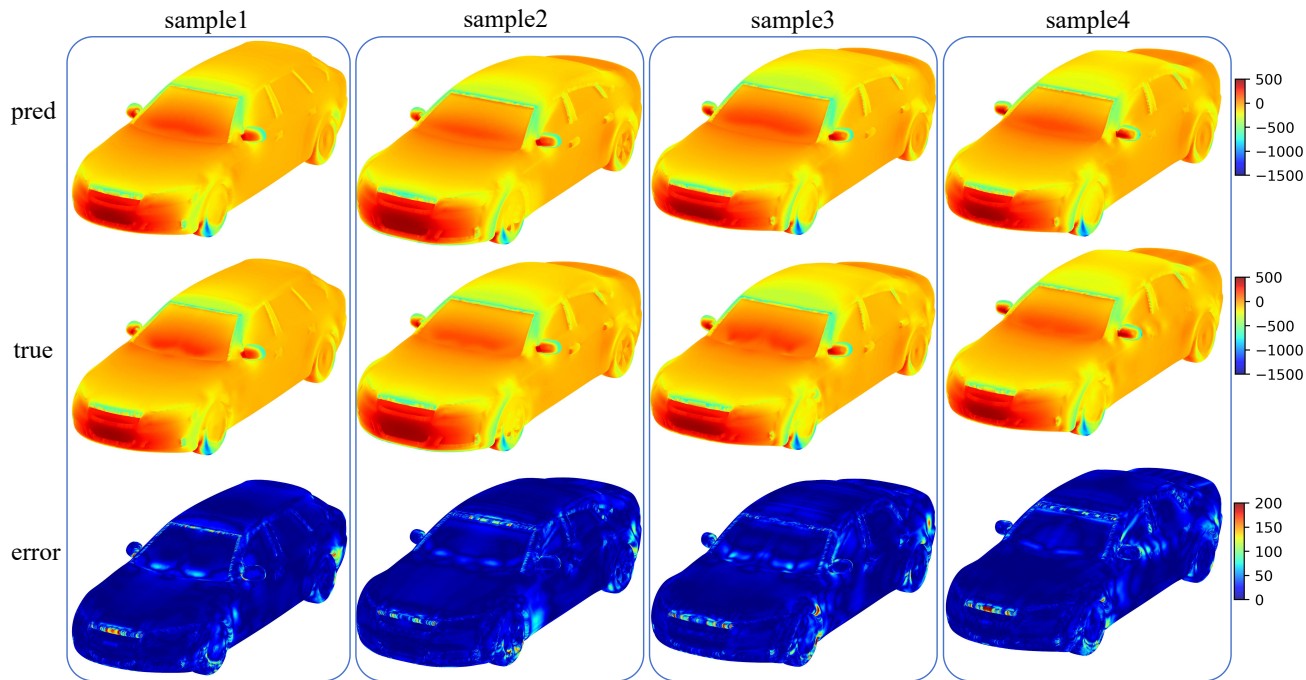

*Figure 7.* The display of prediction results on DrivAerNet++ dataset. The pred represents the predicted pressure fields, the true denotes the ground truth, and the error stands for the absolute error of the prediction fluid fields.

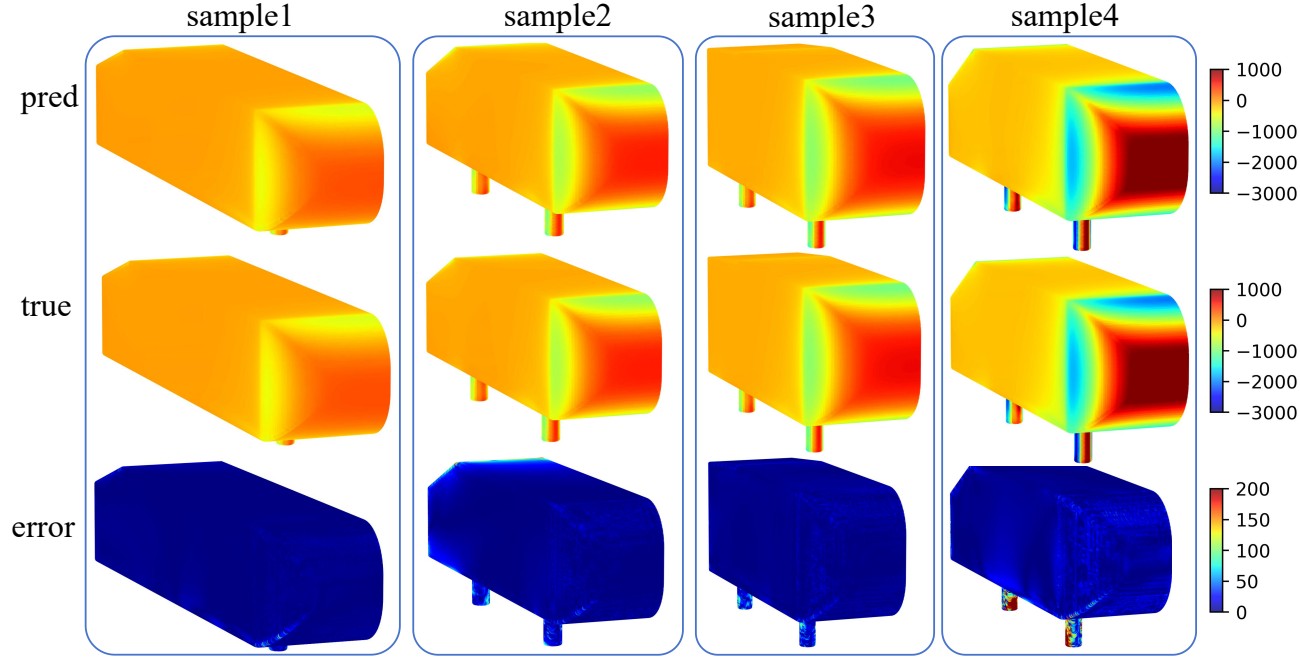

*Figure 8.* The display of prediction results on Ahmed body dataset. The pred represents the predicted pressure fields, the true denotes the ground truth, and the error stands for the absolute error of the prediction fluid fields.

Similarly, noticeable errors appear in the downstream velocity field due to the highly complex turbulent dynamics in the wake region.

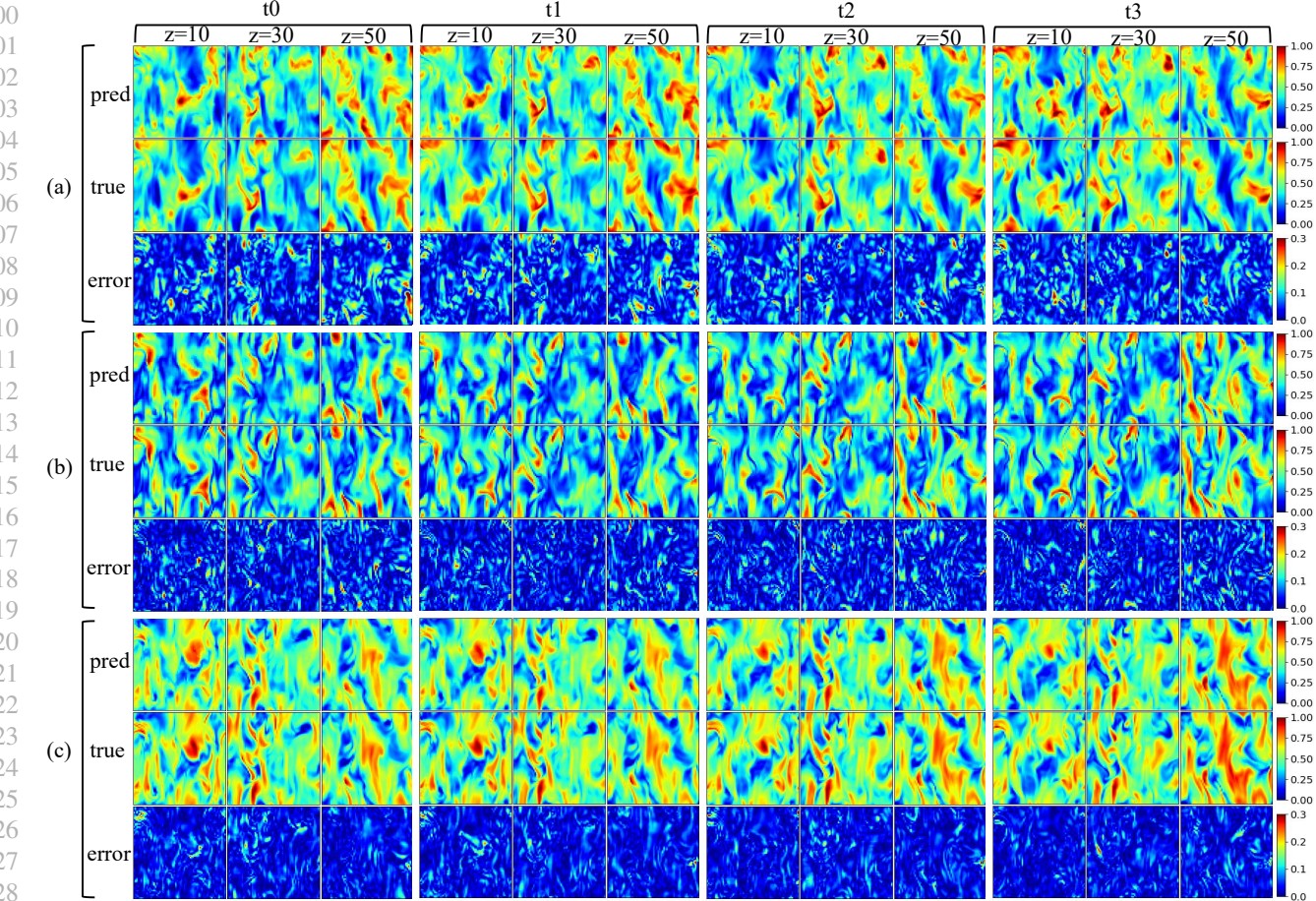

*Figure 9.* The display of prediction results on Magnetohydrodynamics dataset. (a) Density field. (b) Velocity field (vector magnitude). (c) Magnetic field (vector magnitude). z10, z30, and z50 represent the cross-sectional views at positions 10, 30, and 50 along the z-axis, respectively. t0, t1, t2, and t3 represent four consecutive time steps. pred, true, and error represent the prediction, ground truth, and error field, respectively.

## K. Visualization of key flow field regions of MNO and baseline

In this section, we aim to zoom in some local flow fields to observe key regions where the model predictions fail and conduct a detailed discussion and analysis. The DrivAerNet++ benchmark has the highest point-cloud resolution and complex geometries; therefore, we select this dataset for localized visualization.

By comparing the overall prediction errors, the key regions we select include the demarcation line area between the front end and the chassis of the car, the transition area between the rear end and the chassis of the car, the door handle area of the car, the rearview mirror area, and the wheel area. Figure 11 shows the prediction results of MNO and the baseline (Transolver). The baseline model consistently performs worse than our MNO method in these key regions, especially in small-scale areas with abrupt geometric changes. As shown in Figure 11 (d), the rearview mirror area occupies an extremely small proportion of the surface but involves complex geometric deformations, particularly at the connection part between the mirror and the car body. The prediction error of the MNO model in this region is consistently lower than that of the baseline model, which sufficiently demonstrates that MNO's multi-scale strategy possesses a stronger capability for analyzing fine-grained boundary variations in flow fields.

Moreover, Figure 11 reveals a common challenge: all models perform not well around the wheel regions, particularly behind the wheels. Although MNO shows improvement in error metrics compared to the baseline, the performance remain limited in the wheel regions. The interior space of the wheel compartment is narrow and contains numerous components, which collectively form a multitude of intricate gaps and cavity structures. This generates highly disordered turbulent wakes behind the wheels that exceed the fitting capacity of current neural operator models. Consequently, the wheel region will be a key

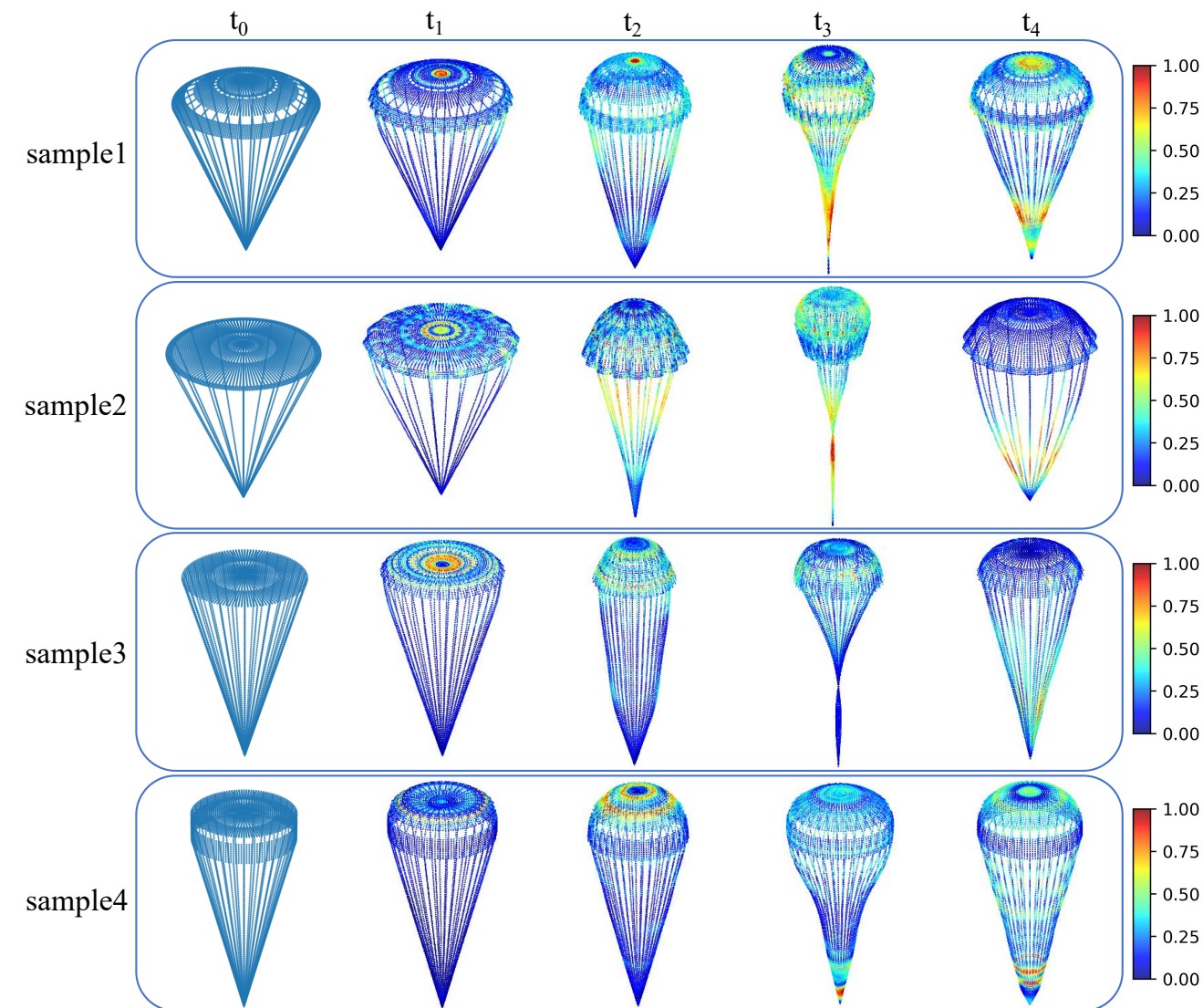

*Figure 10.* The display of prediction results on Parachute dataset. $t0$ is the initial shape of the parachute in the air, while $t1$, $t2$, $t3$, and $t4$ stand for the shape changes of the parachute over 4 time steps. The color of the point cloud represents the prediction error amplitude of displacement fields.

focus for future model improvements.

## L. The Evaluation Metrics

For the quantitative evaluation of point cloud predicion algorithms, this study employs the following two widely used metrics: Relative L2 Error (RL2) and Mean Absolute Error (MAE). Both metrics are calculated based on point-to-point correspondence between the predicted point cloud and the true point cloud, requiring that the point clouds be precisely aligned and point correspondences established prior to evaluation.

### L.1. The Relative L2 Error

The RL2 measures the normalized Euclidean distance discrepancy of the predicted point cloud as a whole relative to the true point cloud. It is defined as follows:

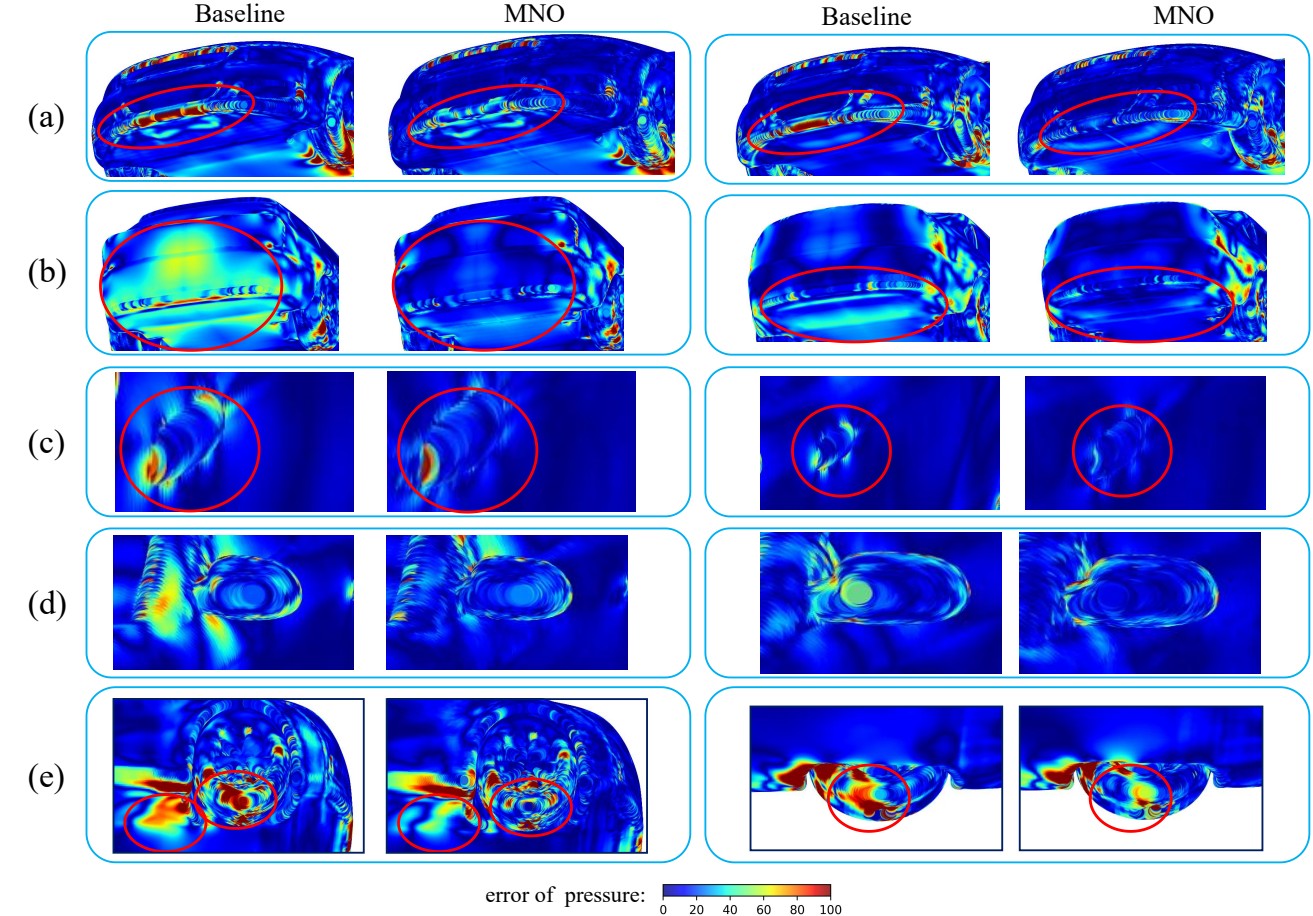

*Figure 11.* Display of local key regions of flow field. (a) Demarcation line area between the front end and the chassis of the car, (b) Transition area between the rear end and the chassis of the car, (c) Door handle area of the car, (d) Rearview mirror area, (e) Wheel area.

$$RL2 = \frac{\|\hat{Y} - Y\|_2}{\|Y\|_2} = \frac{\sqrt{\sum_{i=1}^{N} \|\hat{y}_i - y_i\|_2}}{\sqrt{\sum_{i=1}^{N} \|y_i\|_2}}, \tag{8}$$

where $Y = \{y_1, y_2, \ldots, y_N\}$ is the true point cloud, $\hat{Y} = \{\hat{y}_1, \hat{y}_2, \ldots, \hat{y}_N\}$ is the predicted point cloud, $N$ is the number of points, $\|\cdot\|_2$ represents the L2 norm.

A smaller RL2 value indicates lower relative error between the predicted point cloud and the true point cloud at the overall level, reflecting higher prediction accuracy. By using the norm of the true point cloud as the denominator, this metric achieves scale invariance, enabling robust performance comparisons across different scales or datasets.

**L.2. The Mean Absolute Error**

MAE measures the mean of the absolute deviations between the predicted point cloud and the true point cloud on a point-wise basis. It is defined as follows:

$$MAE = \frac{1}{N} \sum_{i=1}^{N} \|\hat{y}_i - y_i\|, \tag{9}$$

where $Y = \{y_1, y_2, \ldots, y_N\}$ is the true point cloud, $\hat{Y} = \{\hat{y}_1, \hat{y}_2, \ldots, \hat{y}_N\}$ is the predicted point cloud, $N$ is the number of points, $\| \cdot \|_1$ represents the L1 norm.

A smaller MAE value indicates that the predicted point cloud aligns more closely with the ground truth along each coordinate axis, reflecting higher point-wise accuracy. Unlike Mean Squared Error (MSE), MAE is less sensitive to outliers (individual points with large errors), providing a more robust estimate of the average deviation.

The combined use of RL2 and MAE enables a more comprehensive evaluation of point cloud reconstruction algorithm performance: RL2 focuses on the fidelity of global, while MAE assesses localized accuracy. Lower values for both metrics collectively indicate superior reconstruction quality.

## M. LLMs Polishing

The manuscript was polished using Large Language Models (LLMs) of DeepSeek-R1 and ChatGPT-4.0 to improve clarity, grammar, and academic style. The authors rigorously reviewed and edited all AI-generated content to ensure accuracy and consistency with the original scientific intent. The intellectual contributions remain entirely human.

