# OpenReview forum: "MNO: Multiscale Neural Operator for 3D Computational Fluid Dynamics"
_ICML.cc/2026/Conference — Submitted to ICML 2026_

### Official Review · Reviewer_RYsE · 2026-02-19

**Soundness:** 3
**Presentation:** 2
**Significance:** 3
**Originality:** 2
**Overall Recommendation:** 4
**Confidence:** 4

**Summary:**

The paper proposes a new neural operator method for computational fluid dynamics on 3D point clouds. The method consists of three parallel modules: a global attention module that captures long-range dependencies via a low-rank projection, a local attention module that models interactions between neighboring points using differential attention over a shared kNN graph, and a micro attention module for fine-grained point-wise details. The method is evaluated on five diverse benchmarks covering both steady-state and unsteady flow scenarios, where MNO consistently outperforms baselines. The paper demonstrates that these gains stem from the multiscale design itself rather than increased model size, with MNO achieving competitive performance at considerably fewer parameters than the best baselines.

**Compliance With Llm Reviewing Policy:**

Affirmed.

**Final Justification:**

The authors addressed my main concerns: adding Transolver and AB-UPT baselines, providing equal-compute comparisons that confirm architectural gains over scale, and clarifying the Global Module's distinction from Transolver's attention. This justifies raising my score to 4.

However, I stop short of 5 because MNO's inference time is 3–6× slower than key baselines, and I got the feeling that the authors wanted to hide it from the first submission. Nevertheless, i think this paper is a solid contribution to the field.

**Key Questions For Authors:**

- Is the computational cost reported in Appendix D calculated with or without the kNN graph construction time? This matters for a fair comparison with methods that recompute neighborhood graphs across layers.
- Can you comment on the architectural difference between your global attention module and the physics-based attention used in Transolver? The paper would benefit from a more explicit comparison of these two compression strategies.
- Given the geometric generalization limitations shown in Appendix I, what is the path forward for applying MNO to more diverse geometries beyond automotive shapes?

**Limitations:**

yes

**Strengths And Weaknesses:**

## Strength:
- The three modules are thoroughly explained with clear mathematical formulations, and the architectural motivation is well-grounded in the physics of multiscale flow.
- While the individual components are not entirely novel, their combination and tight coupling within each block is well-motivated and the contribution is meaningful.
- The shared kNN graph across all blocks is an elegant design choice that avoids repeated neighbor searches
- Ablation studies are comprehensive, covering module combinations, depth, low-rank dimension M, and neighborhood size k.
## Weaknesses:
- **Missing baselines across tables:** Transolver appears in Table 2 but is absent from Table 1, and conversely PCNO appears in Table 1 but is absent from Table 2. The authors should either include all competitive baselines consistently across all benchmarks or provide a clear justification for omissions.
- **Missing recent baselines:** The comparison omits AB-UPT [1], a recent strong baseline for neural CFD surrogates on automotive aerodynamics. Given the overlap with the DrivAerNet++ and Ahmed body benchmarks, this comparison seems important for establishing state-of-the-art claims.
- **Ablation presentation:** The "Performance of independent modules" paragraph is hard to read and digest. Table 3 contains too many rows and the accompanying text is repetitive. A cleaner format like showing the full model first, then systematically removing one module at a time, would be clearer and more consistent with how ablations are typically presented in the literature.
- **Missing inference time comparisons:** Table 6 only compares inference time against AMG, leaving all other baselines unaccounted for. A comprehensive inference time comparison across all baselines would substantially strengthen the efficiency claims.
- **Minor presentation issues:** Several small but noticeable presentation issues should be corrected before publication:
	- The citation at L19 appears to be incorrectly formatted
	- There is a spurious opening parenthesis in the subscript at L235
	- Figure 1(c) contains a typo — "Sotfmax" should read "Softmax" — which is particularly visible given its placement in the main architecture figure
	- Table 1 and Table 2 appear in noticeably different font sizes despite being directly adjacent, which looks inconsistent and may confuse readers comparing results across the two tables


[1] Alkin et. al., AB-UPT: Scaling Neural CFD Surrogates for High-Fidelity Automotive Aerodynamics Simulations via Anchored-Branched Universal Physics Transformers, TMLR 2025

---

> ### Author Rebuttal · Authors · 2026-03-31
>
> > **Q1**: Transolver baseline supplementation in Table 1.
>
> We added Transolver results on Ahmed Body and Parachute benchmarks (see Table 1 bellow). MNO outperforms Transolver by 13% and 15.8% respectively.
>
> The PCNO codebase lacks implementations on benchmarks of Table 2, and their official code is tightly coupled, making adaptation to other benchmarks difficult, so we omitted. We will continue our baseline reproduction of other methods and complete the comparative experiments in the final version.
>
> **Table 1 (part):** Transolver baseline supplement.
> |            |\|Ahmed body |           |\|Parachute  |   |   |   |  | |
> |---------|---------|---------|---------|---------|---------|-------|-------|--------|
> | Methods    |$RL2_p$   | $MAE_p$    |\|$RL2_{x1}$ | $RL2_{x2}$ | $RL2_{x3}$ | $RL2_{x4}$ | $RL2_x$ | $MAE_x$  |
> | **Transolver**|0.0540 | 8.5383     |\|0.0302     | 0.0229     | 0.0312     | 0.0460     | 0.0316  | 0.0098   |
> | MNO       |0.0468     | 7.0465     |\|0.0216     | 0.0164     | 0.0259     | 0.0418     | 0.0266  | 0.0081   |
>
>
>
> > **Q2**: Comparison with AB-UPT.
>
> Thank you for highlighting this important baseline. AB-UPT is indeed a strong recent work for automotive aerodynamics, published 3 months before our submission. We added comparison experiments (Table R4.1): MNO's relative L2 error is 5.6% lower than AB-UPT, validating our method effectiveness.
>
> AB-UPT targets steady-state fields and does not support time-evolution modeling (e.g., MHD and parachute benchmarks). In contrast, MNO's unified multi-scale architecture naturally adapts to both steady and transient tasks, demonstrating stronger task generalization.
>
> Table R4.1: Comparison with AB-UPT on DrivAerNet++.
> | Methods | Time    | Param  | RL2-p  | MAE-p    |
> |---------|---------|--------|--------|----------|
> | AB-UPT  | 1.3774s | 7.007M | 0.1759 | 15.0892  |
> | MNO     | 3.6256s | 2.039M | 0.1665 | 14.6335  |
>
>
> > **Q4**: Computing cost comparison withh other baselines.
>
> We added computing cost statistics for baselines in Table R4.2.
>
> **Table R4.2(part)**: Cost statistics for all baseline calculations, DrivAerNet++(300k,200samples) as an example. MNO(M=32) included kNN graph construction time.
>
> | Methods    | Flops   | Params | $GPU_i$ | train/epoch | inference/sample  |
> |------------|---------|--------|----|-----|-----|
> | AMG        | 373.8G  | 1.35M  | 9.9G| 16.6h| 299.20s|
> | AB-UPT     |  1.753T | 7.007M | 4.4G| 234s| 1.3621|
> | Transolver | 1.1T    | 3.85M  | 3.1G| 302s| 0.6157s|
> | MNO        | 900.8G  | 2.15M  | 13.4G| 916s| 3.6256s|
>
> **Equal-compute performance comparison:** We provided a new comparison with equal inference time, where we adjusted each model's depth and width (hidden dim) to match inference time.
> Table R4.3 shows MNO's RL2 error remains 5.6%–5.9% lower than AB-UPT, confirming gains stem from architectural design, not higher compute budget.
>
>
> **Table R4.3**: Comparison with AB-UPT at same computational cost on DrivAerNet++.
>
> | Methods | `Time`  | RL2-p  | MAE-p    |
> |---------|---------|--------|----------|
> | AB-UPT  | 3.7298s | 0.1764 | 15.4159  |
> | MNO     | 3.6256s | 0.1665 | 14.6335  |
>
> | Methods | `Param` | RL2-p  | MAE-p    |
> |---------|-------|--------|----------|
> | AB-UPT  | 2.22M | 0.1770 | 15.4186  |
> | MNO     | 2.05M | 0.1665 | 14.6335  |
>
>
> > **Q5**: Spelling, Table, Citation and other formatting issues.
>
> Thank you for those suggestions. We will correct these issues in the manuscript and check for similar formatting problems.
>
>
> > **Q7**: Comparison between Global Module and Translolver.
>
> - **Transolver**: Uses identical weight matrix W(N,M) for both projection and inverse.
> - **Global Module**: Applies **bidirectional Softmax** along N and M dimensions of W, generating two independent but homologous projection matrices P and Q.
>
> This design retains projection consistency while introducing asymmetric mapping capability and enabling modeling of complex physical field dependencies.
>
> **Added comparison tests:** Table R4.4 shows Global Module's RL2 error is ~3.2% lower than Transolver, confirming the bidirectional Softmax mechanism enhances global feature modeling expressiveness while maintaining compute efficiency.
>
> **Table R4.4**: Transolver vs Global (aligned hyperparameters and residual structure) on ShapNet Car.
>
> | Methods      | RL2-p  | MAE-p  | RL2-v  | MAE-v   |
> |--------------|--------|--------|--------|---------|
> | Transolver   | 0.0700 | 1.8151 | 0.0230 | 0.1130  |
> | Global Modue | 0.0662 | 1.5952 | 0.0180 | 0.0837  |
>
>
> > **Q8**: Discussion on the Prospects of Geometric Generalization
>
> We view this as an important direction for future work. We will incorporate shape-adaptive normalization into MNO’s multi-scale architecture to improve its ability to perceive and model unseen geometries. In addition, we plan to build larger and more diverse 3D CFD benchmarks, including geometries such as aircraft, high-speed trains, and buildings, to improve generalization through broader geometric coverage.

---

> > ### Author Rebuttal · Reviewer_RYsE · 2026-04-01
> >
> > I will raise my score to a 4 (weak accept), as all my concerns have been adequately addressed.

---

> > > ### Author Response · Authors · 2026-04-08
> > >
> > > We sincerely thank you for your careful evaluation and constructive suggestions, which have helped improve the quality and clarity of this work.

---

### Official Review · Reviewer_zqQ3 · 2026-03-10

**Soundness:** 2
**Presentation:** 2
**Significance:** 2
**Originality:** 2
**Overall Recommendation:** 4
**Confidence:** 5

**Summary:**

This paper proposes the MNO, an architecture for solving CFD problems directly on 3D unstructured point clouds. MNO decomposes feature learning across three modules within each block: (1) a Global Dimension-Shrinkage Attention module that projects N points into an M-dimensional low-rank subspace for efficient long-range dependency modeling; (2) a Local Graph Attention module that uses a shared kNN graph with differential attention; and (3) a Micro Point-wise Attention module that applies sigmoid gating for per-point feature refinement. MNO is evaluated on five benchmarks (Ahmed Body, Parachute, MHD, ShapeNet Car, DrivAerNet++) that cover steady-state and unsteady flows, with 15K–300K points.

**Compliance With Llm Reviewing Policy:**

Affirmed.

**Final Justification:**

First, I really appreciate the author's hard work for providing the new results in the final time. Although I still have the following concerns, I think the author has proved the effectiveness of their method in some senses, which reaches the borderline level of acceptance by ICML; Therefore, I will raise my score to 4. But the following misunderstandings must be clarified by the author in the CRV.

1. The comparison of the full DrivAerNet++ dataset is not correct. The numbers that the author cites are from the CarBench paper. However, the results of the carbench paper are trained with the fixed 10k downsample resolutions. The results of GAOT are trained with full resolution. I don't know which case the author trained their model on.
2. For the MHD dataset, I agree with the author that this dataset is useful for the Tokamak system. However, this is a very small dataset and is unconvincing to verify the ability of different architectures. Models are prone to overfitting on it. The correct usage of this dataset is to work as a downstream task and verify the transfer abilities of PDE Foundation models.
3. For the scalability experiment, what I am interested in is the scaling properties of MNO with the input resolution and model size. Because Transolver is not a scalable model, this can also be seen in Table R4.2 (from the author's reply to RYsE). Small model parameters will give rise to a very large number of flops but small training throughputs. The hardware compliance is very weak. Because the author's model is built on top of the Transolver and is much heavier, I don't get how they solve this problem and make it faster than Transolver.

**Key Questions For Authors:**

1. Why were only 200 out of 8000+ DrivAerNet++ samples used for training? How does performance scale with training set size?
2. What is the sensitivity of MNO to the fusion strategy? The current design uses the simple addition of the three module outputs. Have other fusion strategies (concatenation, cross-attention ... ) been explored?

**Limitations:**

yes

**Strengths And Weaknesses:**

**Weakness:**
1. Limited Novelty. The proposed components are borrowed from existing work. (1) In my opinon, the global module is exactly the transolver's physics-aware token compression mechanism, which projects N tokens into a compact latent via learned softmax projections, applies attention in the reduced space, and projects back. The paper does not sufficiently acknowledge that this is essentially Transolver's core mechanism but rebranded as "dimension-shrinkage attention." (2) For local module, this is just point transformer's differential attention directly adapted with kNN. (3) Micro module is a standard sigmoid gating, which is an extremely common design.

2. Very small training dataset. Several benchmarks use remarkably small training sets. I noticed that MHD only use 78 training samples. Drivaernet++ only use 200 out of 8000 availabe samples. With such small datasets, it is unclear whether the reported improvements reflect meaningful learning of underlying physics or simply better overfitting capacity. The choice to use only 200/8000+ DrivAerNet++ samples is particularly puzzling and should be justified.

3. The author omits the two important state-of-the-art baselines GAOT (Wen et al., 2025) and UPT (Alkin et al., 2024) that are directly relevant to neural operators on irregular domains. They represent the current state-of-the-art on exactly the benchmarks evaluated in this paper. In particular, GAOT has reported results on DrivAerNet++ that should be directly compared; without this, the claimed improvements on this benchmark are incomplete.

4. Hard to scale. Like Transolver, MNO performs projection (N->M) and inverse-projection (M->N) in every MNO block. This repeated compress-decompress cycle has significant memory overhead and low computational throughput. As the number of MNO blocks increases, this becomes a serious scalability bottleneck. In contrast, methods such as GAOT and UPT perform the projection only once (encode into latent space, process, then decode), making them fundamentally more scalable. The paper does not systematically discuss this scalability limitation. Table 5 shows the computational costs of MNO's own module configurations, but critically lacks a comparison of training memory and wall-clock time against key baselines like Transolver, LNO, and GAOT under the same hardware conditions. From Table 6, it can be observed that on DrivAerNet++ (300K points), MNO's training time per epoch is 0.39 hours. While this is much faster than AMG but much slower than other SOTA models as I know.

---

> ### Author Rebuttal · Authors · 2026-03-31
>
> > **Q1**: Novelty of the three sub modules.
>
> The overall three-scale design, which captures global trends, local dynamic interactions, and fine-grained details at different spatial scales, is novel compared with prior multi-resolution pooling approaches such as PointNet++, GAOT, and Point Transformer. The ablation results in Table 3 and Figure 3 confirm that each module combination improves performance, while the full three-module design achieves the best results. For example, on the challenging magnetohydrodynamics benchmark, MNO reduces the error by about 50% (Table 2), substantially outperforming single-scale variants.
>
> We admit that each of the three submodules (global, local, and micro) is inspired by prior works, including Transolver, LNO, and graph attention methods, but each contains important structural differences and corresponding performance gains.
>
> Specifically, the Global Module differs from Transolver in that it applies **bidirectional Softmax** along the N and M dimensions to construct the P and Q mappings, rather than using a fully shared projection. This improves the flexibility of the compression and reconstruction process. Table R3.1 shows that this design yields a 3.2% error reduction compared with Transolver.
>
> The Local Module is inspired by Point Transformer, but is constructed within a unified kNN graph space (see Figure.1(a,b,d) for details). This allows the differential features in the Local Module to complement the global features from the Global Module and the absolute features from the Micro Module.
>
> **Table R3.1**: Transolver vs Global (aligned hyperparameters and residual structure) on ShapNet Car.
> | Methods      | RL2-p  | MAE-p  | RL2-v  | MAE-v   |
> |--------------|--------|--------|--------|---------|
> | Transolver   | 0.0700 | 1.8151 | 0.0230 | 0.1130  |
> | **Global Modue** | 0.0662 | 1.5952 | 0.0180 | 0.0837  |
>
>
>
> > **Q2**: Training sample size
>
> - **MHD**: The original benchmark setting of *The Well* uses the same 78 training samples. These 78 samples correspond to 78 time-series segments, each containing 100 frames, which is equivalent to roughly 7,800 effective temporal samples.
>
> - **DrivAerNet++**: We followed the setup of Transolver++  (190 training samples and 10 test samples), but further increased the test set to 50 samples.The full dataset of 8,000 samples occupies about 200 GB, whereas 200 samples require only about 5 GB.
>
> We also provide an experiment with more sample, increasing samples from 200 to 500 improves MNO by 4.5%(Table R3.2).
>
> In addition, this small-sample setting is also closer to many real-world scientific and engineering scenarios, and therefore better tests whether a model has effective physical inductive biases rather than relying primarily on data scale. In addition, the ablation trends in Table 3 are consistent with those on the other four benchmarks, which does not indicate overfitting. We will clarify these points more explicitly in the revision.
>
>
>
> >**Q3**: New baselines
>
> The GAOT was published 4 months before our submission, so we prioritized reproducing it, while the experiments for UPT are still running. Table R3.2 shows that MNO outperforms GAOT by 28.4% (200 samples) and 22.1% (500 samples), demonstrating strong data efficiency and generalization.
>
> **Table R3.2**: Comparison with GAOT on DrivAerNet++.
>
> | Methods(train:test)|RL2-p| MAE-p|
> | ---- | ------ | ------- |
> | GAOT (200:50) | 0.2327 | 19.5216 |
> | MNO (200:50) | 0.1665 | 14.6335 |
> | GAOT (500:00)| 0.2042 | 16.2393 |
> | MNO (500:100)| 0.1590 | 13.3450 |
>
> > Q4: Scalability and computational cost.
>
> **Cost Comparison:** Table R2.1 (in Q1 of jbDC response) presents computational cost comparisons on the same hardware. MNO reduces time cost by 90% compared to the latest graph based SOTA (AMG). Other Transformer baselines are generally faster because graph architectures like MNO and AMG lack optimization for current GPU hardware.
>
> Comparison under same computational cost: For fairer comparison, we provide **a new experiment under same inference time constraints**. In Table R1.1 (in Q2 of 2QwY response), MNO still leads by 15.4% (same inference time) , confirming gains stem from architectural design, not higher compute budget.
>
>
> > Q5: Fusion strategy
>
> Thanks for pointing out this extension. We tested concatenation approach shown in  Table R3.3. It yields no significant gain.
>
> **Table R3.3**: Fusion strategy experiment on ShapeNetCar.
>
> | Methods     | RL2-p  | MAE-p  | RL2-v  | MAE-v   |
> |-------------|--------|--------|--------|---------|
> | Cat Fusion  | 0.0619 | 1.5256 | 0.0206 | 0.1064  |
> | Plus Fusion | 0.0597 | 1.3803 | 0.0180 | 0.0857  |
>
> Cross attention incurs $O(N^2)$ complexity and high overhead, so it is not our primary choice.

---

> > ### Author Rebuttal · Reviewer_zqQ3 · 2026-04-03
> >
> > Thanks to the author's clarification and the new experiments. But I still think this paper is below the acceptance bar. The following are my reasons:
> > 1. The mentioned novelty by the author remains very weak. First, the author confirms my point that each module is borrowed from the prior work. The claimed "bidirectional softmax" is a minor engineering adjustment from Transolver, not a conceptual contribution, and 3.2% error reduction is very marginal and could easily fall into the noise on such small datasets. Furthermore, the overall three-scale parallel design is just a packaging strategy for existing methods.
> > 2. The dataset size issue is not resolved. The author mentioned that the MHD justification is 78 samples × 100 frames = 7800 effective samples. However, this is highly misleading. Temporally adjacent frames are highly correlated. For DrivAerNet++, the author says "we followed Transolver++'s setup". This just shifts the attention rather than addressing my concern. Btw, the relative L2 error for pressure is much better than the baselines (which were trained with 5000 samples) in the drivernet++ leaderboard. I am very skeptical about the results. The fact that 200 -> 500 samples improves MNO by 4.5% really raises my question: what happens at 5000? Btw, the author mentions "this small-sample setting is also closer to many real-world scientific and engineering scenarios." Btw, Drivaernet++ is not a high-resolution real 3d scenariso. If the author wants to fight for this point, they should test DrivaerML rather than DrivaerNet++.
> > 3. My scalability concerns are not solved. Comparing only with AMG is cherry-picking. The selling point is for 3D CFD. If the author cannot verify the scalability of the proposed method, the entire narrative falls apart.
> >
> > I will maintain my overall score of 3.

---

> > > ### Author Response · Authors · 2026-04-08
> > >
> > > We appreciate the reviewer’s continued concerns.
> > >
> > > >Q1: The novelty remains weak.
> > >
> > > Our main novelty claim rests on the **three-scale coupled design in a unified graph space**, not on modification of submodules alone. Existing operator models typically emphasize either single-scale global modeling (e.g., Transolver, LNO) or multi-scale hierarchies based on pooling/resampling (e.g., PointNet++, GAOT). In contrast, MNO keeps all three scales active within each block, without explicit resampling or cross-scale alignment. The ablation study in Table 3 shows that removing any branch consistently degrades performance, and that the full three-scale block is required to achieve the best results.
> > >
> > > As for the submodules, the three-scale components are improved upon prior works, rather than identical copies of existing structures. The “bidirectional softmax”  yields steady improvement (though minor) without extra cost.
> > >
> > > Moreover, we introduce a unified graph-space architecture, which enables implicit alignment in a shared point-graph space and avoids the resampling and explicit alignment used in conventional multi-scale methods. The Local module, within a unified kNN graph,  uses differential attention for medium-scale modeling, while the Micro module restores point-level absolute information lost in differencing, which is critical for high-frequency details. These contributions were also acknowledged by other reviewers.
> > >
> > >
> > >
> > >
> > >
> > >
> > > >Q2:The dataset size issue is not resolved.
> > >
> > > **DrivAerNet++ dataset:** The training using full  DrivAerNet++ has now been completed (Table R3.4). Performance improves with more samples. Compared to GAOT, it achieves a 12.9% improvement.
> > >
> > >
> > >
> > > **Table R3.4:**  DrivAerNet++. * is from the official leaderboard or original paper.
> > >
> > > | Methods            | Train Samples | Test Samples | RL2-p      | MAE-p   |
> > > | ------------------ | ------------- | ------------ | ---------- | ------- |
> > > | GAOT               | 200           | 50           | 0.2327     | 19.5216 |
> > > | MNO                | 200           | 50           | 0.1665     | 14.6335 |
> > > | GAOT               | 500           | 100          | 0.2042     | 16.2393 |
> > > | MNO                | 500           | 100          | **0.1590** | 13.3450 |
> > > | Point Transformer* | 5819          | 1154         | 0.1909     | -       |
> > > | Transolver*        | 5819          | 1154         | 0.1503     | -       |
> > > | Transolver++*      | 5819          | 1154         | 0.1573     | -       |
> > > | GAOT*              | 5819          | 1154         | **0.1570** | 12.900  |
> > > | **MNO**            | 5819          | 1154         | **0.1367** | 11.4597 |
> > >
> > >
> > >
> > > The initial 200-sample setting was chosen to align with the Transolver++ experimental protocol (Transolver++, ICML 2025). Due to time limits, DrivAerML will be included in future work.
> > >
> > >
> > >
> > > **MHD dataset:** We strictly follow the official split in *The Well* [1]: 78 training and 20 testing trajectories, each containing 100 frames of $64^3$ data.
> > >
> > > Within each trajectory, the frames are temporally correlated. This is similar to weather datasets, e.g. ERA5, which are still useful and influential cases in AI for weather forecasting. For this MHD dataset, the task is to predict solar wind dynamics, which also has potential application on modeling controlled fusion systems such as Tokamaks. Therefore, we believe it is still a worthwhile benchmark to include.
> > >
> > > [1] The well: a large-scale collection of diverse physics simulations for machine learning. NIPS 2024.
> > >
> > >
> > >
> > >
> > > >Q3 Scalability concerns are not solved.
> > >
> > > First, in our previous response we added an **equal-inference-time comparison** (Table R2.2 in reponse to Review jbDC), where all baselines were re-tuned to match MNO’s wall-clock inference time, and  MNO still achieves the best accuracy on ShapeNet Car.
> > >
> > >
> > >
> > > Second, to directly address the concern about repeated projections, we added a **single-projection variant**, **MNO-OneProj** (Table R3.5). It uses a supernode single projection architecture similar to UPT, where MNO blocks are embedded in the projected latent space to replace stacked transformer blocks. MNO-OneProj get best accuray, and 4X faster, half GPU memory usage than UPT.
> > >
> > >
> > >
> > > **Table R3.5:** Scalability experiment results on DrivAerNet++. (200/50 training/test samples)
> > >
> > > | Methods         | Time    | Param  | $GPU_i$ | $RL2_p$ | $MAE_p$ |
> > > | --------------- | ------- | ------ | ------- | ------- | ------- |
> > > | GAOT            | 0.9311s | 11.24M | 1.9G    | 0.2327  | 19.5216 |
> > > | UPT             | 2.3939s | 11.01M | 4.3G    | 0.1911  | 21.8339 |
> > > | **MNO-OneProj** | 0.4382s | 3.83M  | 2.5G    | 0.1835  | 16.0611 |
> > > | MNO             | 3.6256s | 2.05M  | 13.4G   | 0.1665  | 14.6335 |
> > >
> > >
> > >
> > > Tthe proposed multi-scale block is **compatible with more scalable single-projection pipelines**. In practice, **full MNO** provides the best accuracy, while **MNO-OneProj** offers a stronger efficiency–accuracy trade-off.

---

### Official Review · Reviewer_jbDC · 2026-03-10

**Soundness:** 4
**Presentation:** 4
**Significance:** 3
**Originality:** 3
**Overall Recommendation:** 5
**Confidence:** 4

**Summary:**

A novel architecture, the Multiscale Neural Operator (MNO), is proposed for solving three-dimensional flow modeling problems. The model is distinguished by its ability to process unstructured point clouds by employing enhanced mechanisms of global, local, and point-wise attention. The experiments cover a wide range of benchmarks, varying flow parameters, media, and object shapes. The results demonstrate a significant reduction in error compared to state-of-the-art models. The presented ablation study results provide insight into the specific impact of the mechanisms used in the model. Implementation details and code for result reproduction are also provided.

**Compliance With Llm Reviewing Policy:**

Affirmed.

**Final Justification:**

When evaluating the paper, weaknesses were identified in the performance of the presented method compared to others. In their rebuttal, the authors provided new data that addressed this shortcoming. Based on the rebuttal, I am raising the "Soundness" score to 4. I recommend accepting the paper.

**Key Questions For Authors:**

1. Were performance measurements conducted in comparison with other models?
2. A separate model was trained for each benchmark. What are your thoughts on the possibility of training a multi-task model capable of performing across all considered benchmarks? What are the prospects and obstacles to creating such a model?
3. The Related Work section mentions Physics-Informed Neural Networks (PINNs). However, it cites works from 2019-2023. In recent years, numerous new models and methods have emerged in this field. Could you provide more intuition on why the data-driven approach remains preferable for the benchmarks you consider, and what the prospects for PINNs are in this context?

**Limitations:**

yes

**Strengths And Weaknesses:**

Strengths:

1. A well-structured description of the approach and parameters for result reproduction.
2. Extensive validation on benchmarks, demonstrating the advantage of the proposed approach.
3. A detailed ablation study that provides insight into how the mechanisms used in the model and their parameters specifically affect the outcome.

Weaknesses:

For the problems under consideration, not only the error value but also the computational cost plays a role. Model selection typically relies on a trade-off between accuracy and computational speed. Although the authors show a significant error reduction compared to other models, the paper does not provide a comparison of computational costs. Such data would provide more intuition about the scenarios in which the proposed model would prove most beneficial.

---

> ### Author Rebuttal · Authors · 2026-03-31
>
> > **Q1**: The trade-off between computational cost and accuracy
>
>
> We added detailed compute cost statistics for all baselines across benchmarks (Table R2.1) to aid accuracy-speed trade-off decisions.
>
>
> MNO's training/inference time is aboput 90% lower than latest graph-based SOTA (AMG, KDD 2025). Other Transformer baselines are typically faster because graph architectures (MNO, AMG) lack equivalent GPU optimization.
>
>
> We also added performance comparisons under identical compute budgets (Table R2.2). All baselines were re-tuned (depth/hidden dim) to match MNO's inference time. Results show MNO outperforms baselines by 15.4% under equal compute cost.
>
>
>
> **Table R2.1(part)**: Computational costs of baseline models, ShapeNet Car and DrivAerNet++ as examples.
>
> ShapeNet Car(30k):
> | Methods    | Flops  | Params | $GPU_t$ | $GPU_i$ | train time /epoch | inference time /sample  |
> |------------|--------|--------|---------|---------|-------------------|-------------------------|
> | LNO        | 7.7G   | 1.29M  | 1.2G    | 0.7G    | 64s               | 0.0406s                 |
> | AMG        | 40.1G  | 1.35M  | 4.7G    | 1.6G    | 1.0h              | 4.4398s                 |
> | GAOT       | 90.9G  | 11.23M | 2.2G    | 0.8G    | 166s              | 0.0758s                 |
> | Transolver | 125.3G | 3.86M  | 4.3G    | 0.8G    | 142s              | 0.0703s                 |
> | MNO(M=32)  | 147.8G | 2.04M  | 12.7G   | 3.1G    | 208s              | 0.1681s                 |
>
> DrivAerNet++(300k):
> | Methods    | Flops  | Params  | $GPU_t$ | $GPU_i$ | train time /epoch | inference time /sample  |
> |------------|--------|---------|---------|---------|-------------------|-------------------------|
> | LNO        | 69.5G  | 1.29M   | 6.6G    | 2.3G    | 141s              | 0.4151s                 |
> | AMG        | 373.8G | 1.35M   | 37.9G   | 9.9G    | 16.6h             | 299.20s                 |
> | GAOT       | 198.1G | 11.239M | 8.9G    | 6.5G    | 321s              | 0.9288s                 |
> | Transolver | 1.1T   | 3.85M   | 34.0G   | 3.1G    | 302s              | 0.6157s                 |
> | MNO(M=32)  | 900.8G | 2.05M   | 67.9G   | 13.4G   | 916s              | 3.6256s                 |
>
>
>
> **Table R2.2**: Comparison under the same inference time on ShapeNet Car.
> | Methods    | `Time`| RL2-p  | MAE-p  | RL2-v  | MAE-v     |
> |------------|---------|--------|--------|--------|---------|
> | LNO        | 0.1689s | 0.0772 | 1.9846 | 0.0224 | 0.1129  |
> | AMG        | 0.1724s | 0.0851 | 2.4391 | 0.0280 | 0.1552  |
> | GAOT       | 0.1665s | 0.1255 | 3.0847 | 0.0302 | 0.1472  |
> | Transolver | 0.1663s | 0.0706 | 1.7978 | 0.0198 | 0.0949  |
> | **MNO**    | 0.1642s | 0.0597 | 1.3803 | 0.0180 | 0.0857  |
>
>
> > **Q2**: The feasibility, prospects, and obstacles of multitasking models.
>
>
> Thank you for this insightful question.
>
>
> A multi-task MNO is technically feasible. Its encoder-decoder architecture naturally supports multi-task extension: task-specific output heads + task identifiers as conditional inputs. Prospects include: (1) single training covering wind-tunnel, MHD, parachute, etc., reducing deployment/maintenance costs; (2) shared low-level fluid dynamics representations promoting knowledge transfer and few-shot generalization; (3) alignment with industrial "one-model-multi-scenario" needs.
>
>
> Challenges include: heterogeneous prediction targets (pressure/velocity/displacement/MHD fields) with incompatible dimensions/scales; diverse auxiliary feature encodings (boundary conditions, Re, time steps); varying point cloud resolutions requiring dynamic multi-scale attention adaptation; differing physical mechanisms (steady/transient, laminar/turbulent) complicating loss balancing and gradient coordination.
>
>
> > **Q3**: The Prospects of Data Driven and PINN Methods
>
> Traditional PINNs require training a separate network for each instance, leading to high application costs. Some existing benchmarks do not provide complete parameters for generating equations, making it impossible to compute accurate losses.
>
> A current development direction involves combining PINN with Neural Operators, e.g. PINO. In future, by incorporating physical constraints, we aim to significantly enhance NO's zero-shot generalization capabilities in unseen parameter domains, such as high Reynolds number scenarios.

---

> > ### Author Rebuttal · Reviewer_jbDC · 2026-04-03
> >
> > The author’s responses fully address the questions raised in the review.

---

> > > ### Author Response · Authors · 2026-04-08
> > >
> > > We sincerely thank you for your careful evaluation and constructive suggestions, which have helped improve the quality and clarity of this work.

---

### Official Review · Reviewer_2QwY · 2026-03-13

**Soundness:** 3
**Presentation:** 4
**Significance:** 3
**Originality:** 3
**Overall Recommendation:** 5
**Confidence:** 4

**Summary:**

This paper proposes multiscale neural operator (MNO) on 3D computational fluid dynamics on unstructured point clouds, which integrates three different scale attention modules: 1) long-range dependencies with global dimension-shrinkage, 2) neighborhood interactions with local graph, and 3) fine-grained details with micro-point-wise mechanisms.

**Compliance With Llm Reviewing Policy:**

Affirmed.

**Key Questions For Authors:**

1. The paper reports geometric generalization study, but it is unclear how well MNO generalizes to out-of-distribution boundary conditions, or flow regimes varying Reynolds numbers not seen during training. Can the authors share further evaluation on extrapolation performance or robustness?

**Limitations:**

Yes

**Strengths And Weaknesses:**

Strengths
- The work presented on comprehensive problems spanning from wind tunnel to parachute simulations and consistently resulted in superior performance, especially in high turbulent magnetohydrodynamics (MHD) case.
- It also presented thorough ablation study and computational analysis on both training and inference runtime and memory consumptions.

Weaknesses
- While the work has been thoroughly demonstrated, the proposed methodology can be considered as a bit intuitive without significant novelty or originality. However, the comprehensive demonstration and study of the parameters outweighs this.
- Computational cost of the proposed model is considerably high, which was exhibited consistently across all benchmarks.

---

> ### Author Rebuttal · Authors · 2026-03-31
>
> > **Q1**: While the work has been thoroughly demonstrated, the proposed methodology can be considered as a bit intuitive without significant novelty or originality. However, the comprehensive demonstration and study of the parameters outweighs this.
>
> Thank you for recognizing the contribution of proposed model and the validation of the comprehensive experiments. We'd like to provide additional discussion on the 'intuitive' structure of MNO and novelty.
>
> **Three-Scale Modeling:** MNO explicitly targets three physical regimes (global trends, local interactions, fine details) via three modules respectively. Table 3 shows that any module combination boosts performance, confirming they capture underlying physics rather than simple component stacking.
>
> The three (global/local/micro) modules are inspired by and improved from existing works (Transolver,LNO, graph attentions etc).
>
> In particular, on the global module, we apply bidirectional Softmax on N and M dimensions for P, Q mappings, increasing flexibility while maintaining semantic consistency, outperforming Transover by ~3.2% (Table R3.1 in Q1 of zqQ3 response).
>
> On the local graph module, we desined a unified Graph Space Architecture. Unlike traditional multi-resolution pooling (PointNet++, GAOT, Point Transformer), we proposed a shared point-cloud graph structure enabling continuous cross- scale interaction without costy feature alignment.
>
>
>
>
> > **Q2**: Computational cost of the proposed model is high.
>
>
> MNO reduces time cost by 90% compared to the latest graph based SOTA (AMG). We admit that our method as well as other graph based methods, e.g. AMG, is generally slower than pure transformer based method, due to inefficient implementation of graph NN on GPU hardware, though theoretical TFLOPS and parameters is even smaller.
>
> For fairer comparison, we provide a new experiment under **same inference time constraints**. By adjusting the depth/hidden dim, we compare smaller MNO/AMG models with larger Transolver/LNO. All models run at about 0.18 seconds. As Table R1.1 shows, MNO still outperforms the strongest baseline by 15.4%, confirming gains arise from architectural efficiency, not compute budget.
>
> **Table R1.1(part)**: Comparison under the same inference time on ShapeNet Car
> | Methods    | `Time`    | Param   | RL2-p  | MAE-p  | RL2-v  | MAE-v   |
> |------------|---------|---------|--------|--------|--------|---------|
> | LNO        | 0.1689s | 13.44M  | 0.0772 | 1.9846 | 0.0224 | 0.1129  |
> | AMG        | 0.1724s | 124.83K | 0.0851 | 2.4391 | 0.0280 | 0.1552  |
> | GAOT       | 0.1665s | 11.24M  | 0.1255 | 3.0847 | 0.0302 | 0.1472  |
> | Transolver | 0.1663s | 15.62M  | 0.0706 | 1.7978 | 0.0198 | 0.0949  |
> | MNO        | 0.1642s | 2.04M   | 0.0597 | 1.3803 | 0.0180 | 0.0857  |
>
>
> > **Q3**: Reynolds number extrapolation performance.
>
>
> Thank you for this valuable suggestion. We added a Reynolds number extrapolation evaluation on Ahmed Body: trained on 500 samples with Re < 4.2e6, tested on 50 samples with Re > 4.2e6. As Table R1.2 shows, MNO maintains ~0.08 relative L2 error on unseen high-Re conditions, demonstrating the transferability.
>
>
> **Table R1.2**: Reynolds number(Re) extrapolation results on Ahmed body. Random represents random Re mixing experiment.
> | Train (500samples) | Test (50samples) | RL2-p  | MAE-p    |
> |--------------------|------------------|--------|----------|
> | Re<4.2e6          | Re>4.2e6         | 0.0818 | 29.8284  |
> | Random             | Random           | 0.0468 | 7.0465   |
>
>
> Performance drop under extrapolation vs. mixed training stems from two factors: (1) High-Re flows are inertia-dominated, prone to separation, shear-layer instability, and small-scale turbulence. The increased nonlinearity raises prediction difficulty; (2) The extrapolation setting uses only 500 training samples with sparse coverage of high-Re regions, limiting the model's ability to fit extreme flow states.
>
>
> Future work will incorporate physics-informed losses (e.g., mass conservation, momentum residuals) to enhance extrapolation consistency, combined with active learning to strategically sample critical high-Re regions for improved out-of-distribution robustness.

---

> > ### Author Rebuttal · Reviewer_2QwY · 2026-04-02
> >
> > I appreciate authors' diligence in providing additional comments, explanations and additional experimental results. As my original score is to accept, there aren't other follow up questions to alter this decision. However, I do want to raise one concern on the future work to improve the extrapolation capabilities. Depending on the flow regimes, the governing equations will vary, so incorporating physics-informed loss will have to be carefully designed.

---

> > > ### Author Response · Authors · 2026-04-08
> > >
> > > Thank you for the insightful suggestion. We are actively exploring hybrid Neural Operator + Physics-Informed Neural Networks (NO+PINN) approaches, and agree that physics-informed constraints must be carefully designed across different flow regimes; we put it as our near future work, with ongoing progress to be reported soon.

---

### Decision · Program_Chairs · 2026-04-30

**Decision:**

Reject

**Comment:**

This paper proposes "MNO", which combines a global attention block, local kNN graph attention, and aggregating features from two levels. The application scenario is for surrogate modeling in 3D CFD problems with seemingly strong benchmark scores. This paper earned the highest score in my batch so I gave it a second look, I had some concerns over its new contribution over AMG (KDD 2025) on whether this is architecturally new or mainly some hyperparams tweaking from the AMG paper.
This lack of elaboration of the new contribution in neural architecture is also confirmed by reviewers, e.g., the Transolver-style compression, multilevel aggregation (e.g., MgNO paper by He-Liu-Xu ICLR 2024, or GAOT paper by Wen et al. NeurIPS 2025), and (graph) attention-based or Swin-based neural operator. Given the current form, I recommend rejection. I recommend citing those relevant works (multilevel feature aggregation, local-global fine-coarse attention based neural operators), giving details on the architectural improvement from AMG, the scalability concern raised in the review, and re-submit.